# Continental-scale genomic surveillance of *Plasmodium falciparum* malaria across sub-Saharan Africa with rapid nanopore sequencing

In sub-Saharan Africa, continental-scale genomic surveillance of *Plasmodium falciparum* malaria is needed to track the spread of drug and diagnostic resistance, as well as monitor parasite evolutionary responses to vaccine rollout. Yet continental-scale implementation is hindered by a lack of genomic approaches suitable for local laboratories, and the vastness of the continent. Here, we initiate a decentralised scale-up of *P. falciparum* genomic surveillance by locally sequencing and analysing 1065 samples across six African countries in one year. We achieve this with a novel nanopore sequencing protocol that is rapid (~5 hr) and cost-effective (<$25 USD/sample), providing surveillance of antimalarial drug resistance genes, *hrp2/3* deletions, the vaccine target *csp*, and the polymorphic gene *ama1*. We couple this to a laptop-based bioinformatics dashboard that runs offline and displays mapping and variant calling results in real-time. We demonstrate robust sequencing coverage across parasitemia levels and laboratories, accurate identification of antimalarial resistance markers and *hrp2/3* deletions; and, with a novel variant caller, sensitive detection of mutations carried by minor clones. Our approach will accelerate genomic surveillance of *P. falciparum* malaria across sub-Saharan Africa at a time of urgent need.

The endeavour to eliminate *Plasmodium (P.) falciparum* malaria from sub-Saharan Africa—where it causes over half a million deaths annually—relies heavily on accurate diagnosis with rapid diagnostic tests (RDTs) and effective treatment with artemisinin-based combination therapies (ACTs)[1]. *P. falciparum*, however, is evolving to undermine both of these tools. For RDTs, *P. falciparum* strains that have deletions of the *hrp2* and *hrp3* genes evade diagnosis by producing false-negative test results[2–6]. For ACTs, mutations in the *kelch13* gene[7] can cause delayed parasite clearance[8] and, if combined with partner drug resistance, treatment failure[9]. Since at least 2016, these *kelch13* mutations have been spreading in East Africa[10–13], where they have begun to co-circulate with *hrp2/3* deletions[14]. They are now being described in Southern[15] and Central Africa[16,17]. Public health institutions are urgently

exploring a variety of responses, such as switching to alternative RDTs when the *hrp2/3* deletion frequency surpasses a threshold regionally[18]; or deploying multiple first-line therapies (MFT) to mitigate the spread of *kelch13* mutations[19,20]. However, essential to coordinating these responses are high-quality, granular, and timely malaria surveillance data, which are lacking in many parts of Africa.

The need to increase malaria surveillance in Africa could be met using genomics. Genomic methods like amplicon sequencing[21–23] and molecular inversion probes (MIPs)[24,25] can interrogate tens to thousands of *P. falciparum* genes in parallel, enabling reporting on many control-relevant genetic markers at once. This makes them more efficient than conventional molecular methods, which typically only report on one gene or mutation. Moreover, many genomic methods

✉e-mail: hendry@mpiib-berlin.mpg.de

**Table 1 | Amplicons included in NOMADS-Minimal Viable Panel (MVP)**

| No. | Reference ID | Gene name | Amplicon (bp) | Codons spanned | Key regions |
|---|---|---|---|---|---|
| 1 | PF3D7_1133400 | ama1 | 826 | 74–384 | |
| 2 | PF3D7_0709000 | crt | 692 | 14–125 | K76T |
| 3 | PF3D7_0304600 | csp | 1273 | 19–398[a] | RTS,S and R21 epitopes |
| 4 | PF3D7_0417200 | dhfr | 1405 | 1–410 | A16V, N51I, C59R, S108M, I164L |
| 5 | PF3D7_0810800 | dhps | 1327 | 317–707[a] | S436A/F, A437G, K540E, A581G, A613T/S |
| 6 | PF3D7_0831800 | hrp2 | 1296 | 14–306[a] | |
| 7 | PF3D7_1372200 | hrp3 | 1216 | 14–276[a] | |
| 8 | PF3D7_1343700 | kelch13 | 1286 | 383–727[a] | BTB/Propeller |
| 9 | PF3D7_0523000 | mdr1(n) | 935 | 46–245 | N86Y/F, Y184F |
| 10 | PF3D7_0523000 | mdr1(c) | 604 | 968–1278 | S1034C, N1024D, D1246Y |

For *kelch13* all WHO candidate and validate markers are spanned[33]. *mdr1*(n) and *mdr1*(c) indicate N- and C-terminal *mdr1* amplicons respectively. bp, base pairs.
[a]Amplicon spans C-terminal codon.

require just a single dried-blood spot (DBS) as the sample, making them less invasive and labour-intensive than standard surveillance approaches like therapeutic efficacy studies. In principle, these genomics approaches could affordably and routinely generate high-quality malaria surveillance data. But scaling *P. falciparum* genomic surveillance to meet the growing need across sub-Saharan Africa is an immense challenge. First, the region is vast and diverse, spanning over 24 million km² and encompassing 49 countries. Perhaps 50,000–100,000 samples would need to be sequenced annually to have adequate statistical power to detect emerging mutations across the entire region (Supplementary Note 1). Second, access to Next-Generation Sequencing (NGS) platforms is limited in many areas. Where NGS platforms do exist, maintenance and the procurement of reagents are invariably unreliable and expensive. Similarly, the computational infrastructure (and/or internet connectivity) required to process large volumes of genomic data is frequently lacking. As a consequence, most *P. falciparum* genomic data have been generated by shipping samples out of Africa, a slow process that also impedes local capacity development and ownership. More recently, centralised sequencing facilities located within Africa are beginning to generate malaria genomic data domestically and regionally, but questions over capacity and country ownership persist.

The possibility of scaling *P. falciparum* genomic surveillance across sub-Saharan Africa in a decentralised way has been relatively neglected. The decentralised approach would leverage the portable and low-cost MinION device from Oxford Nanopore Technologies (ONT)[26] to conduct genomic surveillance from a network of laboratories that span the continent. This would circumvent the need for large capital investments associated with centralised facilities and Illumina platforms, and instead engage more of the existing funding, infrastructure and human potential latent in the thousands of conventional laboratories across Africa. These laboratories could serve their local geography by interacting closely with nearby clinics and public health officials; while simultaneously coordinating activities and sharing data to improve national and continental understanding. Beneficially, a network of decentralised sequencing laboratories would be more robust to disruption and could grow autonomously and exponentially. Until now, the major drawback for this paradigm has been that nanopore sequencing protocols for *P. falciparum* genomic surveillance, although increasing in number[27–30], are less mature and comprehensive than Illumina-based ones[21–23,31]. For example, state-of-the-art Illumina-based amplicon sequencing protocols[23] concurrently provide information on antimalarial drug resistance, *hrp2/3* deletions, the vaccine target *csp* and multiple genetic diversity markers. In contrast, existing nanopore sequencing protocols typically provide information on only one[27,28] or two[29,30] of these outputs. An exception is the NOMADS16 protocol[32], but its focus on generating long-read

data (3–4 kbp) results in it being substantially less sensitive than short-read approaches.

Here, we have filled this gap by developing a novel nanopore sequencing protocol for *P. falciparum* malaria that is rapid, cost-effective, sensitive, and comprehensive—providing information on a panel of genes associated with antimalarial drug resistance (*crt*, *dhfr*, *dhps*, *kelch13*, *mdr1*), as well as *hrp2/3* deletions, the vaccine target *csp*, and the highly polymorphic gene *ama1*. To enable on-site analysis, we developed a real-time bioinformatics pipeline and dashboard that can run on the same laptop used for sequencing. Using these tools, we locally sequenced and analysed 1065 *P. falciparum* DBS samples from six laboratories spanning sub-Saharan Africa in one year. In addition, we developed a novel variant caller to enable sensitive detection of minor clones in polyclonal *P. falciparum* infections and to validate critical aspects of protocol performance across mock and field samples.

## Results

### A protocol to rapidly scale-up decentralised malaria genomic surveillance

We developed an approach to accelerate and scale decentralised *P. falciparum* genomic surveillance across sub-Saharan Africa, called the NOMADS Minimal Viable Panel (MVP) protocol. The approach includes a novel amplicon panel targeting nine genes of public health relevance (Table 1), as well as a nanopore sequencing protocol optimised to minimise costs, complexity, and time. Starting from extracted DNA, the protocol costs $21 USD/sample, including all plasticware ($5.47 USD/sample, 26.2%) and reagents ($15.45 USD/sample, 78.3%; Supplementary Table 1). All required equipment costs an estimated $12,500 USD (Supplementary Table 2). To objectively measure protocol complexity, we quantified the number of pipetting steps from extracted DNA to sequencing. For a batch of 48 samples, our protocol requires 172 pipetting steps, compared to 292 steps for our previous long-read protocol[32] or over 400 steps for two state-of-the-art Illumina-based protocols[23,31] (Supplementary Table 3). The combined protocol incubation time is ~3.5 h, and after gaining familiarity, from extracted DNA to sequencing takes ~5 h (Fig. 1a, Supplementary Fig. 2).

### A real-time, point-of-use bioinformatics pipeline and dashboard

We developed a real-time, on-site bioinformatics pipeline and analysis dashboard that accompanies the NOMADS-MVP protocol, called *Nomadic* (see the "Methods" section, Fig. 1b). *Nomadic* maps reads, computes quality control statistics for each sample and amplicon, performs preliminary variant calling and annotation, and presents this information in a graphical dashboard while sequencing is ongoing. It allows laboratory scientists to make informed decisions regarding sample quality and amplicon performance, determine when to stop

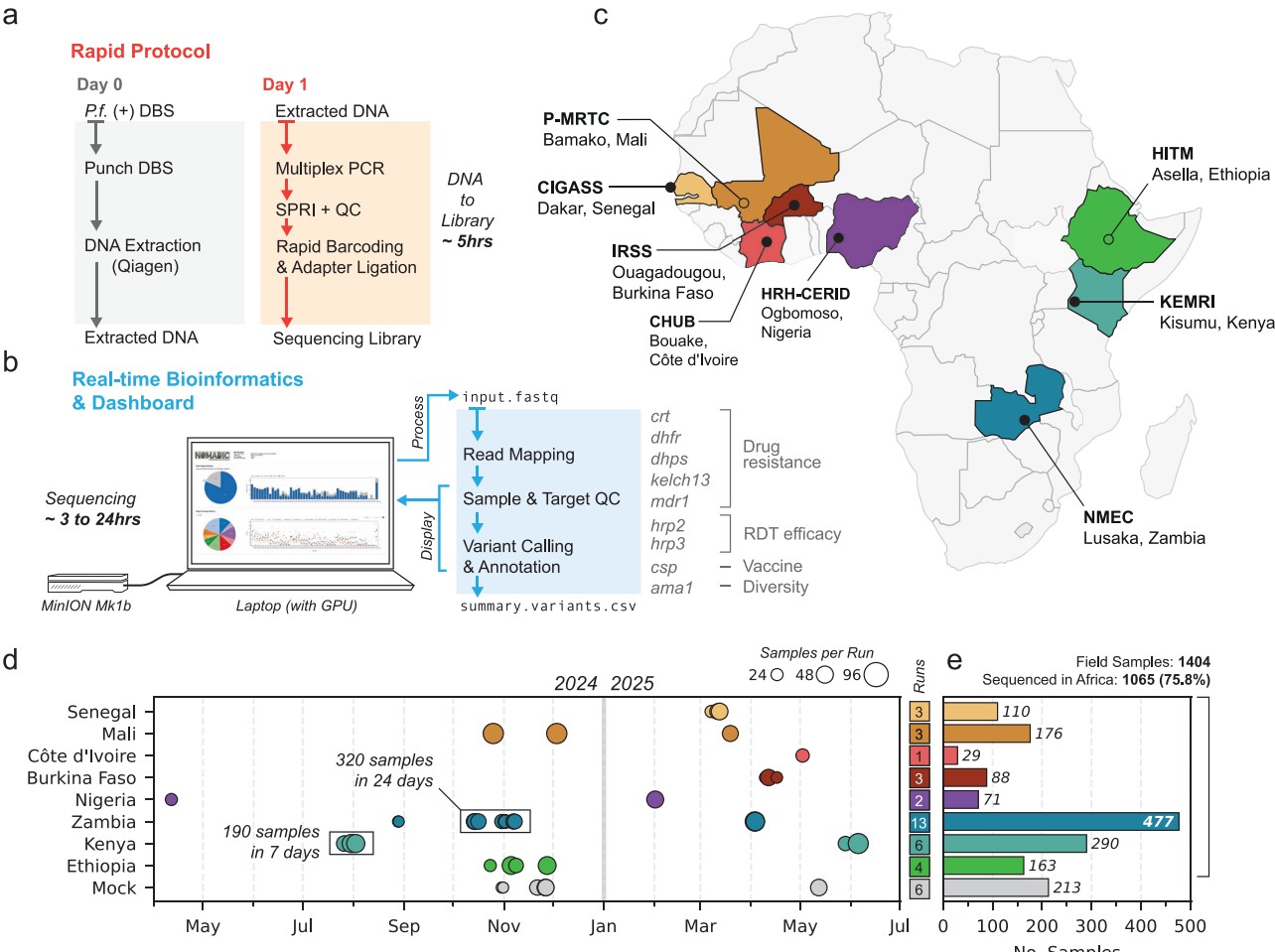

**Fig. 1 | Overview of sequencing approach and implementation across sub-Saharan Africa. a** The laboratory protocol uses *P. falciparum* positive dried-blood spots (DBS) as source material for DNA extraction. From extracted DNA to sequencing takes ~5 h. **b** Data analysis occurs in real-time on a laptop. While sequencing is ongoing, quality control (QC) and variant calling results are displayed on an interactive dashboard. GPU graphics processor unit. Sequencing time depends on flow cell quality and the number of samples. **c** DBS samples were processed from 8 countries across sub-Saharan Africa. For six countries, all sequencing occurred locally (filled circles); for two countries, samples were sequenced internationally (Mali, Ethiopia; open circles). **d** Timeline of sequencing runs. Size of the point indicates the number of samples per run. Total sequencing runs for each country are shown in a box at right. **e** Barplot displaying the total number of samples sequenced per country. Overall, 1065/1404 (75.8%) of samples were processed in Africa. Source data are provided as a Source Data File.

sequencing, and provides an immediate first-look at mutations present across samples. The dashboard is easy to install and is launched with a single command from the terminal, enabling use by scientists without significant bioinformatics training. Beyond what is required to perform real-time basecalling with *MinKNOW* (a laptop with a CUDA-enabled graphics processor unit), *Nomadic* requires no additional computational infrastructure and runs offline. In addition, *Nomadic* produces human-readable summary files that contain all key information about sequencing performance and detected mutations. These files are small enough (<20 Mbp) to be easily shared even over slow internet connections and are sufficient to reopen the graphical dashboard after the experiment is completed. *Nomadic* was used to process all the data described in this manuscript at the point of sequencing.

### Implementation at scale across sub-Saharan Africa

From April 2024 to July 2025, a total of 1404 dried-blood spot (DBS) samples from eight countries across sub-Saharan Africa were sequenced using the NOMADS-MVP protocol (Fig. 1c–e). Of these, 1065 (75.8%) were sequenced locally, with the protocol being established in six countries: Nigeria (April 2024), Kenya (July 2024), Zambia (August 2024), Senegal (March 2025), Burkina Faso (April 2025), and Côte d'Ivoire (May 2025) (Fig. 1c). Overall, 13 local

scientists independently conducted 28 sequencing runs across these six countries (Fig. 1d). The NOMADS-MVP protocol enabled periods of high sample processing throughput. For example, in Kenya two scientists processed 190 samples in 7 days at the end of July 2024 (Fig. 1d); and in late 2024, 320 samples were processed in 24 days by two scientists in Zambia.

Implementation of the protocol was feasible across laboratories that varied considerably with respect to prior sequencing experience and available infrastructure. One team had both Illumina and ONT sequencing experience (CIGASS, Senegal); three teams had previous nanopore sequencing experience (NMEC, Zambia; HRH-CERID, Nigeria; CHUB, Côte d'Ivoire); and for the remainder, our protocol was the first experience with NGS. Infrastructure ranged from well-equipped regional sequencing hubs (CIGASS, Senegal) to small container laboratories with only the essentials for nanopore sequencing (NMEC, Zambia). For five countries, an initial in-person training was conducted by a core team (typically 1–3 weeks in duration) and followed by remote support. In Côte d'Ivoire, the protocol was established locally through a collaboration with the Robert Koch Institute (Berlin, Germany). Overall, these results demonstrate that our approach is amenable to large-scale implementation across a wide range of contexts in sub-Saharan Africa.

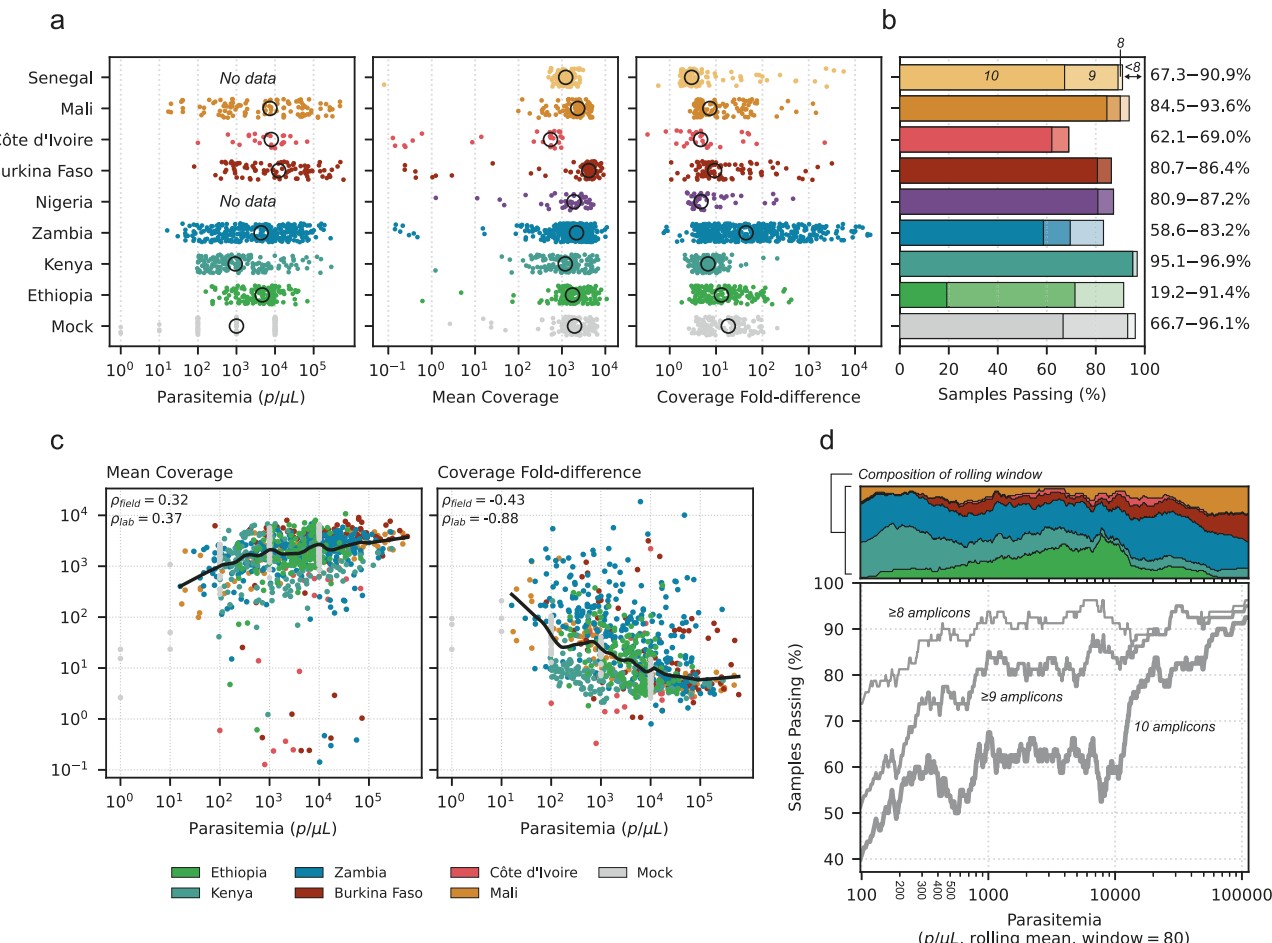

**Fig. 2 | Sequencing coverage across countries and parasitemia levels. a** Strip plots of parasitemia and sequencing coverage data for 1283 samples processed with the NOMADS-MVP protocol, grouped by country (Senegal, 110 samples; Mali, 110; Côte d'Ivoire, 29; Burkina Faso, 88; Nigeria, 47; Zambia, 457; Kenya, 162; Ethiopia, 151; Mock, 129). Each point represents a sample and black circles indicate country-level medians. The left subpanel shows sample parasitemia (parasites/µL); the middle subpanel shows mean per-sample coverage across amplicons; and the right subpanel shows per-sample coverage fold-difference between the least and most abundant amplicon (Methods). All x-axes are on a logarithmic scale. **b** Bar plot showing, for each country, the percentage of samples with >50× coverage for 8 amplicons (lightest shade), 9 amplicons (intermediate), or all 10 amplicons (darkest); bar height indicates the total percentage of samples with >50× coverage for ≥8 amplicons. At right, percentages for 10 amplicons and ≥8 amplicons passing

>50 × coverage are annotated. **c** Scatter plots of parasitemia against mean coverage (left subpanel); and coverage fold-difference (right subpanel). Each point is a sample, coloured by country. Black lines are a locally weighted scatterplot smoothing (LOWESS). Spearman's $\rho$ for the field ($\rho_{field}$) and lab ($\rho_{lab}$) samples is annotated at top left. Both axes use logarithmic scales. **d** Relationship between parasitemia and samples passing (%). Field samples were ordered by parasitemia, and rolling means were computed using windows of 80 samples. Lines represent the percentage of samples with >50× coverage for ≥8 amplicons (thinnest line), ≥9 amplicons (medium), and for all 10 amplicons (thickest). The top subpanel shows the proportion of samples from each country (by colour) across the rolling parasitemia windows. Note that many Ethiopian samples (green) carried *hrp2/3* deletions, causing a decline in the ≥9 amplicon and 10 amplicons lines. Source data are provided as a Source Data File.

## Robust sequencing coverage across countries and parasitemia levels

We examined the sequencing coverage generated using NOMADS-MVP on a set of 1283 samples, comprising 1154 field DBS samples (1154/1283, 89.9%) and 129 mock DBS samples (129/1283, 10.1%), all processed without selective whole-genome amplification (Fig. 2). Parasitemia data were available for 950 samples (950/1283, 74.0%), which together had a median of 3,645 parasites/µL (IQR 631–11,540 parasites/µL). Across all samples, the median per-sample mean coverage was 1881× (IQR 1064–3086×), with 76.3% (881/1154) of field DBS samples and 87.5% (113/129) mock DBS samples achieving a mean coverage of greater than 1000×. The median per-sample fold-difference in coverage was 10.2× (IQR 4.8–53.6×), with samples sequenced in Senegal having the most uniform coverage across amplicons (median 2.97×, IQR 2.14–4.95×, n = 110) and those from Zambia had the least uniform coverage (median 44.4×, IQR 9.5–197.4×, n = 457; caused by lower *dhfr* and *dhps* coverage, Supplementary Fig. 3). Overall, 96.1% (124/129) of

mock DBS samples and 87.9% (1015/1154) of field DBS samples had at least 8 amplicons with greater than 50× coverage; and in 68.2% (88/129) of mock DBS samples and 64.5% of field DBS samples (745/1154) all 10 amplicons exceeded 50× coverage, with part of the reduction due to *hrp2/3* deletions (Fig. 2b).

We examined the effect of parasitemia on sequencing coverage, and observed a moderate positive correlation with per-sample mean coverage for both field DBS samples (Spearman's $\rho$ = 0.32, p < 0.001) and mock DBS samples ($\rho$ = 0.37, p < 0.001). A stronger, negative correlation was observed between parasitemia and the per-sample fold-difference in coverage across amplicons (field DBS: $\rho$ = − 0.43, p < 0.001; mock DBS: $\rho$ = − 0.88, p < 0.001); driven primarily by reduced coverage over the two longest amplicons at low parasitemia levels (*dhfr* and *dhps*, Supplementary Fig. 3). For field DBS samples, at 1000 parasites/µL >90% of samples had at least 8 amplicons with 50× coverage, compared to ~75% of samples at 100 parasites/µL. The mock DBS samples generated from laboratory strains performed

consistently well even at 100 parasites/µL (Supplementary Fig. 4). One explanation is that while mock DBS samples recapitulate absolute parasitemia, they represent a best-case scenario in terms of DNA quality, which may be challenging to achieve in the field. These data suggest that for field samples, a pragmatic inclusion threshold for sequencing would fall between ≥100–1000 parasites/µL (corresponding to ~0.002–0.02% infected red blood cells; or 2.5–25 parasites per 200 white blood cells), but this will depend on DNA quality and should be established on a per-study basis. Relative to parasitemia, only small effects on amplicon coverage were observed between clinical versus asymptomatic, or local versus internationally sequenced samples (Supplementary Fig. 5). Overall, these results demonstrate that NOMADS-MVP can generate robust sequencing coverage across a diversity of sample sets and laboratories in sub-Saharan Africa.

### Sensitive and precise SNP calling in polyclonal infections

Sensitive and specific detection of single-nucleotide polymorphisms (SNPs) is critical for accurate antimalarial drug resistance prediction from known genetic markers. Yet in polyclonal *P. falciparum* infections, SNPs carried by low-frequency minor clones are difficult to detect, especially from nanopore data. To address this, we developed a novel variant caller named *Delve* (Methods), which models the within-sample alternative allele frequency (WSAF) of each SNP as a continuous proportion and enables SNP detection from minor clones.

To assess the performance of *Delve*, we created 45 mock DBS samples from the *P. falciparum* laboratory strains 3D7, Dd2, and HB3. These included clonal samples (3D7, Dd2, or HB3) and two-strain mixtures (3D7 and Dd2, 3D7 and HB3, or HB3 and Dd2) at different minor clone proportions (20%, 10%, 5% and 2.5%) and parasitemia levels (10,000, 1000, and 100 parasites/µL). We sequenced the 45 mock DBS samples in triplicate on three separate R10.4.1 flow cells. For true heterozygous SNPs, the WSAFs centred around the minor clone proportion, confirming that the mock DBS were created accurately (Fig. 3a, Supplementary Table 4, Supplementary Note 2).

Next, we called SNPs using both *Delve* and *bcftools*. We compared their results to the true SNPs for each mock DBS sample and evaluated performance by calculating precision and recall of homozygous alternative and heterozygous SNPs (Methods). The mean recall of homozygous alternative SNPs was perfect (100%) for both tools. *Delve* had a higher recall of heterozygous SNPs than *bcftools* in all 108 polyclonal mock DBS samples (Fig. 3b). In samples with minor clones at 10%, the recall of heterozygous SNPs was high at all parasitemia levels with *Delve* (10,000 parasites/µL: 99.3%; 1000: 99.6%; 100: 94.5%) but low with *bcftools* (10,000 parasites/µL: 15.8%; 1000: 20.3%; 100: 35.6%). Overall, the data indicated that the approximate limit-of-detection (LoD) for minor clones is 5% at 10,000 parasites/µL (mean recall 99.2%, *n* = 9 samples), 5% at 1000 parasites/µL (mean recall 94.4%, *n* = 9) and 10% at 100 parasites/µL (mean recall 94.5%, *n* = 9).

The mean precision across all mock DBS samples and replicates with 1000 parasites/µL or greater (*n* = 90) was very high: 99.9% for *Delve* and 99.8% for *bcftools*. At 100 parasites/µL, however, the mean precision of *Delve* was lower than *bcftools* (94.5% versus 99.7%, *n* = 45 samples), with *Delve* calling a mean of 0.95 false-positive SNPs per sample (range 0–4). All of the false-positives had a low WSAF (median 2.8%, IQR 2.1–3.9%, *n* = 45), with 88% (40/45) having a WSAF <5% (Fig. 3c; Supplementary Note 3). In addition, the false-positives had much weaker statistical evidence in support of a SNP (likelihood ratio test [LRT] statistic, median 24.5, IQR 13.4–43.9) than did true-positive heterozygous SNPs (LRT statistic, median 606.1, IQR 156.4–2266.4); it would, therefore, be possible to filter them, but with a penalty to recall (Supplementary Fig. 7). Significantly, across all mock DBS samples, when the WSAF was between 5% and 95%, *Delve* detected all heterozygous SNPs and had a precision of 99.7% (2195/2200 SNPs). A focused analysis of WHO-defined antimalarial resistance markers[33]

demonstrated very high accuracy in resistance marker calling (Supplementary Note 4).

Last, we used downsampling to explore the relationship between sequencing coverage and the SNP calling performance of *Delve* (Fig. 3d). The recall of true homozygous alternative SNPs was perfect (100%, *n* = 1911 samples) and the mean precision was very high (99.1%, *n* = 1911 samples) for all samples in this analysis (except for clonal 3D7 samples, Supplementary Note 5). The recall of heterozygous SNPs was high, with only 200× coverage for 10% minor clones (10,000 parasites/µL: 95.5%, *n* = 21 samples; 1000: 94.2%, *n* = 21); whereas with 5% minor clones, it reached 89.6 at 10,000 parasites/µL and 84.4% at 1000 parasites/µL. For comparison, across all field DBS samples passing quality control, 82.1% (9362/11,390) of amplicons exceeded 500× coverage, suggesting that detection of minor clones as low as 5% in field samples should be routinely achievable.

### Reliable and streamlined detection of *hrp2/3* deletions

To evaluate the ability of NOMADS-MVP to detect *hrp2* and *hrp3* deletions, we sequenced a set of 149 mock DBS samples collected from malaria patients in central Ethiopia in four MinION sequencing runs. We included mock DBS samples with different *hrp2/3* deletion genotypes as well as *P. falciparum*-negative samples in each run, and inferred *hrp2/3* deletions using a previously described statistical model[32] that leverages coverage across the amplicons (see the "Methods" section).

For all the mock DBS samples, NOMADS-MVP correctly estimated the *hrp2/3* deletion genotype (Fig. 4a). Mean amplicon coverage tended to decline with parasitemia, and low levels of sequencing contamination were evident in some *P. falciparum*-negative controls. Notwithstanding, *hrp2/3* deletions produced a very clear signal; for both *hrp2* and *hrp3*, the difference in coverage between mock samples with and without deletions consistently exceeded 100-fold (Fig. 4c, d; Supplementary Table 6).

For the field DBS samples from Ethiopia, we compared *hrp2/3* deletions inferred by NOMADS-MVP with results from conventional PCRs targeting *hrp2* and *hrp3*, which had previously been performed in duplicate[34]. Of the 149 field DBS samples, 138 (92.6%) passed quality control with NOMADS-MVP. Among these, the joint *hrp2/3* deletion genotypes from NOMADS-MVP matched those from the conventional PCRs in 136 samples (98.6%; Fig. 4d). Investigation of the two discrepancies suggested they were caused by polyclonal *P. falciparum* infections where not all of the clones harboured *hrp2* or *hrp3* deletions (Supplementary Note 6); in these cases NOMADS-MVP detected the deletion, where conventional PCR did not. Overall, these results demonstrate that NOMADS-MVP is a reliable and streamlined method that enables simultaneous detection of *hrp2* and *hrp3* deletions, yielding results that align closely with conventional PCR assays currently in widespread use.

## Discussion

Genomic surveillance of *P. falciparum* malaria can generate valuable public health data, but implementation across sub-Saharan Africa has been limited by a lack of sequencing approaches suitable for most local laboratories. Here, we described the development of a novel nanopore sequencing protocol for *P. falciparum* genomic surveillance, which targets ten genomic regions, collectively providing information on a panel of antimalarial drug resistance-associated genes, *hrp2/3* deletions, the vaccine target *csp*, and the highly diverse gene *ama1*. In one year, we implemented the protocol in Senegal, Burkina Faso, Côte d'Ivoire, Nigeria, Zambia and Kenya—locally sequencing and analysing 1065 DBS samples—and demonstrating the feasibility of continental-scale decentralised *P. falciparum* genomic surveillance.

After genomic data are generated, difficulties performing bioinformatic analysis in sub-Saharan Africa often create another barrier to timely surveillance. We addressed this by developing a real-time

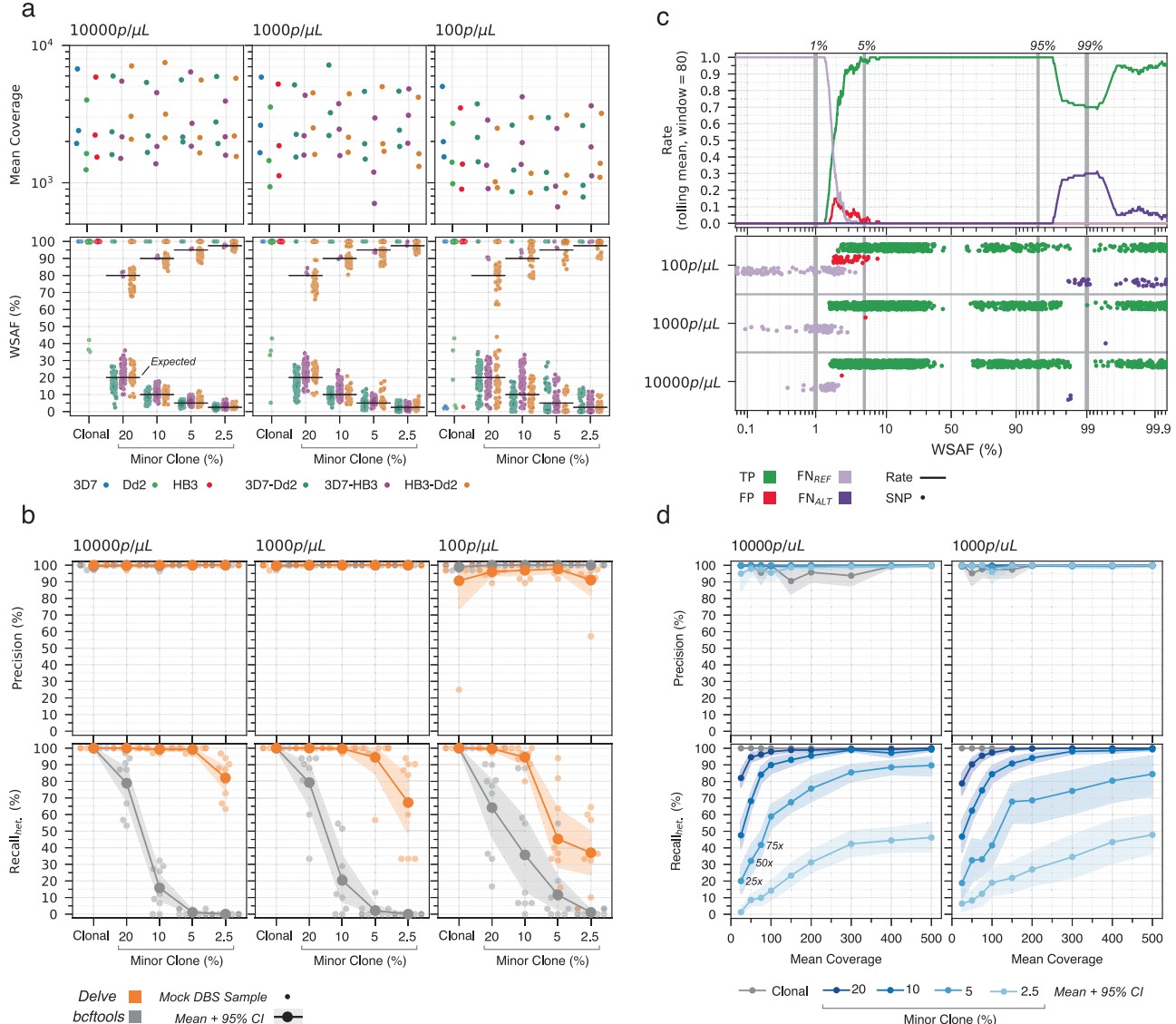

**Fig. 3 | SNP calling performance in clonal and polyclonal mock dried blood spot samples.** A total of 45 mock dried blood spot (DBS) samples were sequenced in triplicate. **a** Strip plots showing the mean coverage per mock DBS sample (top), and within-sample alternative allele frequency (WSAF) for every observed single-nucleotide polymorphism (SNP) (bottom). Mock DBS samples are defined by their constituent strain(s) (colour) and minor clone proportion (*x*-axis). In the bottom subpanels, the horizontal bars indicate the expected WSAF given the minor clone proportion. Note that due to duplication of *mdr1*, the clonal Dd2 samples have one heterozygous site. **b** Same arrangement as (**a**), but the top row of subpanels shows the SNP calling precision, and the bottom shows the recall of true heterozygous SNPs (Recall_het), for both *Delve* (orange) and *bcftools* (grey). Small points represent individual mock DBS samples; the line and shaded area represent the mean and 95% confidence intervals (CI), calculated by bootstrapping. The recall of homozygous alternative SNPs was perfect for both tools. **c** Relationship between observed WSAF

and SNP calling outcome for *Delve*. Each true or called SNP was classified as either a true positive (TP, green), false positive (FP, red), or false negative, in which a heterozygous SNP was called homozygous reference (FN_REF, light purple) or homozygous alternative (FN_ALT, dark purple). SNPs were ordered by WSAF, and a rolling rate for each SNP category was computed using windows of 80 SNPs (top subpanel). Individual SNPs are shown in a strip plot (bottom subpanel) grouped by parasitemia (*y*-axis). The majority of false-positive SNPs have a WSAF of <5%. **d** Effect of downsampling on precision (top) and the recall of heteroyzgous SNPs (bottom). Mock DBS samples with 10,000 and 1000 parasites/µL were downsampled in triplicate to different coverage levels (*x*-axis). Colour indicates clonal (grey) or minor clone proportion (shades of blue). Lines and shaded areas represent the mean and bootstrapped 95% confidence intervals. Source data are provided as a Source Data File.

bioinformatics dashboard called *Nomadic*, which has two main advantages. First, it can be run with minimal bioinformatics expertise, enabling a wide range of users to independently and immediately derive insights from their data. Second, it runs offline on the same laptop used for sequencing, making it affordable and invulnerable to the internet connectivity issues, which can bottleneck workflows reliant on transferring data to remote servers[30]. While analysis occurs on-site and raw data can remain local, *Nomadic* produces compact summary files that can be easily shared. These summary files are useful for collaborative troubleshooting and, in the future, could facilitate

regional data integration when decentralised genomic surveillance becomes widespread.

Identifying variants associated with antimalarial drug resistance is a core aim of genomic surveillance. Yet in polyclonal *P. falciparum* infections, variants carried by minor clones are often missed by standard variant callers, which often assume a diploid organism. We addressed this by developing a novel variant caller, named *Delve*, and demonstrated its ability to recall SNPs carried by minor clones down to 5% frequency while maintaining a high precision. We highlight that *Delve* is a relatively simple variant caller, calling only biallelic SNPs,

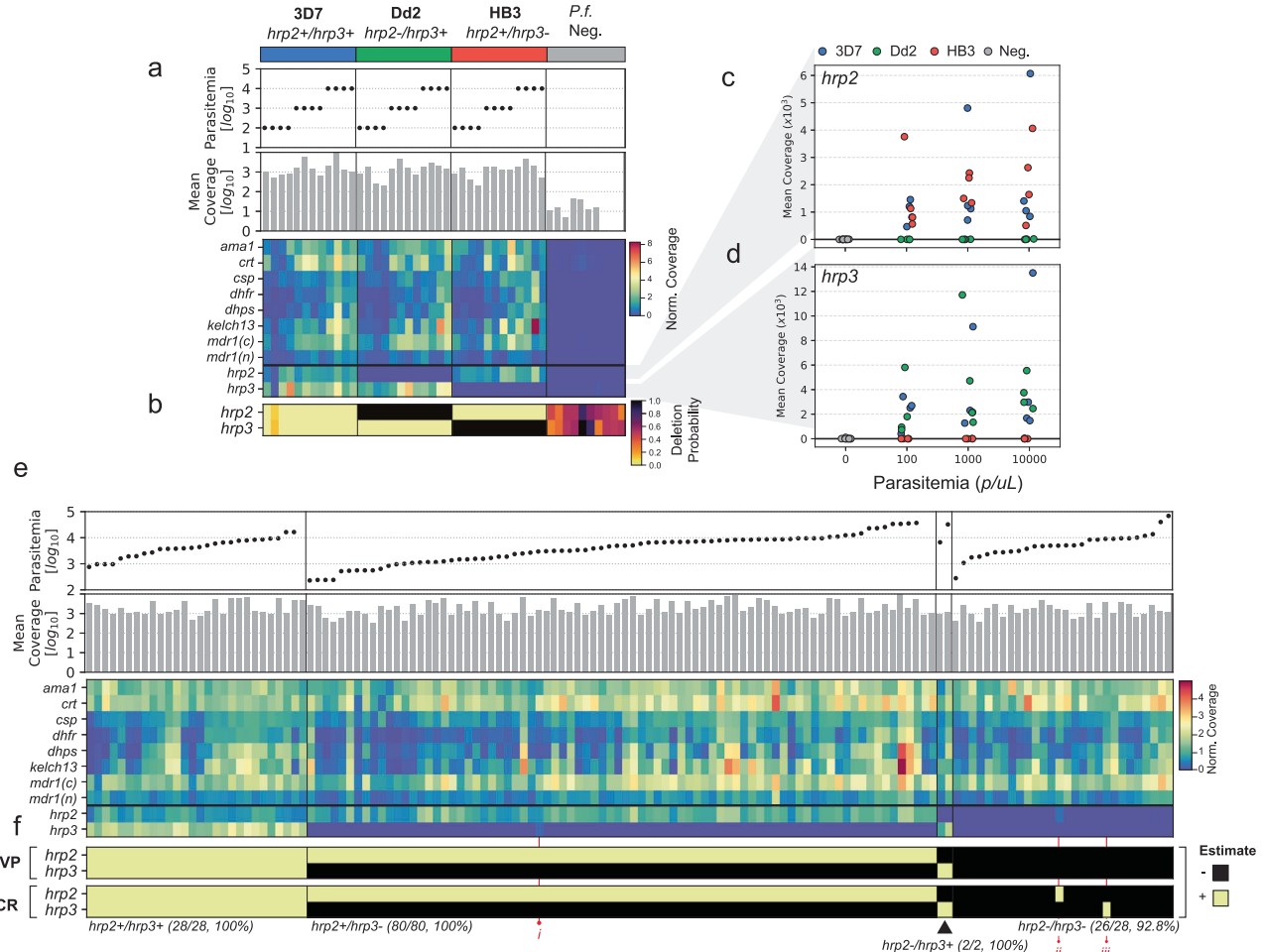

**Fig. 4 | Validation of NOMADS-MVP *hrp2/3* deletion detection on mock and field dried blood spot samples. a** Data for 36 mock dried blood spot (DBS) samples of different *hrp2/3* genotypes and mock *P.f.* samples, which contain only human DNA (see the "Methods" section). From top to bottom, subpanels show: lab strains used in mock samples; scatter plot of parasitemia; bar plot of mean coverage; heatmap with rows indicating different amplicons in NOMADS-MVP, columns indicating samples, and colour indicating mean coverage, normalised to the mean coverage for the experiment. **b** Heatmap showing probability of deletion for *hrp2* and *hrp3* from the statistical model. Note uncertainty (i.e., probability in 0.2–0.8) for negative controls, due to low or absence of coverage. **c** Scatter plot showing mean coverage over *hrp2* for all mock samples. **d** Same as **c**, but for *hrp3*. **e** Same as **a**, but showing data for 149 Ethiopian DBS samples. Vertical lines delineate *hrp2/3* deletion status as estimated by NOMADS-MVP. **f** Estimated *hrp2/3* deletion status by NOMADS-MVP (top subpanel) and the conventional PCR-based assays (bottom). Samples marked with numerals in red font indicate discrepancies (ii, iii) or residual coverage (i). Source data are provided as a Source Data File.

and refraining from recalibrating base quality scores, sharing information across variant sites, or using prior information about population-level allele frequencies. It is possible that more complex somatic or viral variant callers (e.g., *Strelka2*[35], *LoFreq*[36]) could be calibrated to *P. falciparum* nanopore sequencing data and outperform *Delve*, although this has not been explored. We also highlight that haplotype calling tools originally designed for Illumina data, such as *DADA2*[37] and *SeekDeep*[38], have been increasingly applied to nanopore data[30] to detect low-frequency minor clones. Haplotypes are more informative than SNPs and, when combined with long reads, can enable surveillance of important but repetitive genomic regions, such as the NANP-NVDP repeat region of *csp*. However, existing haplotype calling tools require unfragmented reads derived from complete amplicons, making them incompatible with the transposase-based rapid barcoding kits that enable much faster and simpler laboratory protocols for nanopore sequencing. Rather than inferring haplotypes directly, the long reads produced by NOMADS-MVP could be used to physically phase SNPs called by *Delve* with a tool like *WhatsHap*[39] (see Supplementary Note 7). Many salient drug-resistance phenotypes depend on combinations of SNPs, and this approach will enable NOMADS-MVP to predict them accurately even in polyclonal *P. falciparum* infections.

Beyond drug resistance, NOMADS-MVP includes the *hrp2* and *hrp3* genes to enable detection of deletions that can cause false-negative RDT results. Surveillance of the *hrp2/3* genes is important, but challenging[18]; for all existing assays, deletions are detected as a negative signal (e.g., absence of an agarose gel band or sequencing coverage), which can occur for technical reasons unrelated to genotype[40]. In addition, for many assays, polyclonal infections in which only some clones carry deletions can yield ambiguous results. For this surveillance, the World Health Organization (WHO) recommends dual-method testing (with an *hrp2*-based RDT and microscopy or an *ldh*-based RDT) followed by molecular confirmation with either conventional PCR, multiplex real-time PCR (RT-PCR), digital droplet PCR (ddPCR) or DNA sequencing[40]. Similar to available RT-PCR[41] and ddPCR[42] assays, NOMADS-MVP is internally controlled (by the other non-*hrp* amplicons in the multiplex PCR), which can reduce false-positive deletion calls caused by technical failure. A potential advantage of NOMADS-MVP is that it bundles *hrp2/3* and drug resistance surveillance into a single assay, which could enable more rapid and

cost-effective studies in regions where both types of surveillance are needed.

Our protocol does have limitations that make it less well-suited for some malaria genomic surveillance activities. The longer amplicons (500–1500 bp) generated allow contiguous coverage of target genes and simpler multiplexing, but also require samples to have higher parasitemia and/or DNA quality for robust amplification. For clinical cases of malaria, we have demonstrated robust performance with NOMADS-MVP across a wide range of studies. However, for asymptomatic or submicroscopic cases, or deployment in pre-elimination settings, short-read sequencing approaches that leverage smaller amplicons[23,31] are likely to have more consistent performance and less dependence on DNA quality; although we note that several studies included here contained asymptomatic cases and performed well (including dry-season samples from Mali and community-collected samples from Zambia). Similarly, for studies distinguishing *P. falciparum* recrudescence from reinfection, short-read approaches may enable more sensitive minor clone detection[23,30,31]. Future work directly comparing to common Illumina-based protocols[23,31] would help elucidate any potential differences in sensitivity and SNP calling performance, and should include emergent *kelch13* mutations (e.g., R561H, A675V, R622I) in East Africa[9]. Beyond sensitivity, our assay lacks some outputs generated by other assays, such as between-sample relatedness estimates. Relatedness and identity-by-descent (IBD) are best estimated by leveraging a larger number of unlinked variant sites (10s–1000s) spread across the genome. Highly multiplexed short-read amplicon sequencing[21–23], MIPs[24,25], SNP barcodes[43] or whole-genome sequencing (WGS) methods are better suited to make these estimates for studies that require them. Accurate COI estimation will be possible leveraging the *ama1* and *csp* amplicons, and work to provide this output is underway.

The limitations of the NOMADS-MVP protocol are counterbalanced by its ability to detect important threats to malaria control and case management with simplicity and rapidity − thereby meeting an urgent public health need. It is feasible to sequence a moderate-sized batch of samples (24–64) and use *Nomadic* to generate results within a single day, and with substantially less laboratory manipulation than other approaches. The protocol can be easily implemented even in small laboratories with limited infrastructure, internet connectivity, or prior sequencing experience. Because it takes only one or two days to complete, scientists both process sample sets faster and, in the context of training, gain mastery faster, repeating the protocol and accruing experience quickly. Reducing the set of required reagents makes procurement for the protocol easier, simplifies troubleshooting, and ultimately makes outputs more consistent. Moreover, with a rapid protocol, the consequences of a disruption (e.g., by a power outage) are less profound, and scheduling sequencing around other activities is easier. Altogether, these features will accelerate decentralised *P. falciparum* genomic surveillance across sub-Saharan Africa. For widespread and routine implementation, the greatest remaining challenge is no longer technical feasibility, but rather translation from research into public health. This will require guideline development by the WHO, Africa CDC and other authorities; industry-supported standardisation and procurement processes; quality-assurance frameworks; and integration into malaria control programmes and public health laboratories.

## Methods

### Sample collection and ethics

All blood samples were collected from patients with *P. falciparum* malaria, with informed consent from the patient or from a parent or guardian. In all cases, consent allowed for the samples to be used for purposes such as this study. This study was conducted using the collected samples only; there was no human subject contact.

All studies received ethical approval from an appropriate research ethics committee: in Senegal, ethical approval for the SEN19/49 study was granted by the Comité National d'Ethique pour la Recherche en Santé (000317/MSAS/CNERS/SP); in Torodo, Mali, ethical approval for a cohort study in 2022 was granted by the Ethics Committee of Charité (EA2/264/21) and the Faculty of Medicine, Pharmacy and Odontostomatology (FMPOS) at the University of Bamako (N°2022/20/CE/USTTB/24.01.22); in Burkina Faso, ethical approval for the AMTIP study was granted by the Comité d'Ethique Institutionnel/Institut National de Santé Publique (2023-10/MSHP/SG/INSP/CEI); in Côte d'Ivoire, ethical approval for the study "Surveillance génomique de *Plasmodium falciparum* aux CTA à Bouaké, Côte d'ivoire" was granted by the Direction Médicale et Scientifique du aux Centre Hospitalier et Universitaire (192MSHPCMU/CHU-B/DG/DMS/ONAR/24) and by the Ethikkomission der Charité Universitätsmedizin (EA2/171/24); in Nigeria, ethical approval for samples collected in 2023–2024 was granted by the Ethical Committee of Ladoke Akintola University of Technology Teaching Hospital, Ogbomoso (LTH/OGB/EC/2022/304); in Zambia, ethical approval for the 2024 Malaria Indicator Survey (MIS2024) and the 2024–2025 *hrp2/3* Surveillance Study was granted by the Research Ethics Committee at the University of Zambia (5055-20241) and the Tropical Diseases Research Centre Ethics Review Committee (TRC/C4//03/2024), respectively; in Kenya, ethical approval for samples collected in the ATSB study was granted by the KEMRI Scientific and Ethics Review Unit (KEMRI/SERU/CGHR/368/4189; CDC Project ID 0900f3eb81d7ec3c and 0900f3eb82546323); in Ethiopia, ethical approval for the ARSUNA study was granted by the Arsi University institutional review board (AU/HSC/ST-129/5494) and the Federal Democratic Republic of Ethiopia, Ministry of Education (17/256/476/24).

### Mock sample creation

**Making DBS from culture.** For validation of our protocol, we created mock *P. falciparum* positive dried-blood spots (DBS) as follows. We cultured the *P. falciparum* laboratory strains 3D7, Dd2 and HB3 at 5% haematocrit in commercially available red blood cells (RBCs) obtained from DRK-Blutspendedienst Nord-Ost gGmbH[44]. We brought each culture to ~5% parasitemia and then synchronised to ring-stage parasites using 5% Sorbitol (PanReac AppliChem, #A2222). After synchronisation, three technicians independently measured the parasitemia of each culture by microscopy and used a Neubauer Counting Chamber to determine the number of RBCs per microlitre. We used the average of the parasitemia and RBCs/μL measurements to standardise each culture to 100,000 parasites/μL at 50% haematocrit. We performed 10-fold serial dilutions of the 100,000 parasites/μL stocks in 80 μL of whole human blood to produce clonal mock samples down to 1 parasites/μL. To explore sequencing performance on polyclonal infections with low-frequency minor clones, we created mock samples containing two laboratory strains at different proportions. In particular, we combined clonal 10,000 parasites/μL dilutions of two strains to create 80:20 mixtures in 120 μL, and then performed two-fold serial dilutions into the major strain to produce 90:10, 95:5, and 97.5:2.5 mixtures. These were further serially diluted into whole human blood to produce 1000 parasites/μL and 100 parasites/μL parasitemia mixtures. To produce the DBS, we created five 20 μL spots for each mock sample on an individual filter paper (Whatman, #10531018) and dried them overnight at room temperature (Supplementary Fig. 4a). The DBS were stored at −20 °C in individual plastic bags with a desiccant until use.

**DNA extraction from DBS.** For each sample, we used a single (6 mm) punch from the DBS for DNA extraction. DNA was extracted by using QIAamp DNA Mini Kit (Qiagen, #51306) according to the manufacturer's instructions with the following modifications: elution was performed by using 2 × 40 μL of nuclease-free water heated to 50 °C. Each elution was incubated for 3 min at room temperature.

**Mock samples containing *kelch13* mutations.** To interrogate artemisinin-resistance mutations, we ordered *P. falciparum* genomic DNA for Cambodian field-derived strains IPC 5202 (*kelch13* R539T), IPC 4912 (*kelch13* I543T), IPC 3445 (*kelch13* C580Y)[45] from BEI resources (www.beiresources.org). We created 10,000 parasites/µL in vitro DNA mixtures by diluting these stocks to 0.25 ng/µL in 25 ng/µL human genomic DNA (Roche #11691112001).

## NOMADS-MVP multiplex PCR

**Primer design.** We designed primers for a novel amplicon panel using our open-source multiplex PCR primer design software, *Multiply*[32] (https://github.com/JasonAHendry/multiply). We targeted nine genes that would jointly provide information on antimalarial drug resistance, *hrp2/3* deletions, vaccine target evolution and complexity of infection (COI) (Table 1). Each gene was targeted with a single amplicon, except *mdr1* for which separate N- and C-terminal amplicons were designed. For the ten targets, *Multiply* produced a total of 453 candidate primer pairs (median 47 per target); across which it identified 4096 potential offtarget-binding sites, 77 primers overlapping common variants (>5% minor allele frequency (MAF) in any population defined in the Pf6 Project[46] or drug resistance-associated codons), and 2028 high-affinity primer dimers. The greedy search algorithm minimised these factors to select a set of 3 candidate multiplexes (from a theoretical 22 thousand trillion possible combinations) for the ten targets. After pilot sequencing experiments, we selected the multiplex PCR with the highest coverage over all ten targets as our final panel (Supplementary Fig. 1d). The included amplicons vary in size from 604 to 1405 bp and amplify a combined 10,860 bp in a single reaction. A total of 45 validated and candidate World Health Organization (WHO) antimalarial drug resistance-associated single-nucleotide polymorphisms (SNPs) are genotyped (Table 1). The amplicons targeting *hrp2* and *hrp3* both have reverse primers annealing in exon 1, and forward primers upstream of exon 2, which enables detection of partial or complete hrp gene deletions[47]. The *csp* amplicon spans from codon 19 to the end of the gene, which includes the entire central repeat region and C-terminal domain used in the RTS,S/AS01 and R21/Matrix-M vaccines[48,49]. A high diversity, 826 bp window of *apical membrane antigen 1* (*ama1*) is covered to support COI estimation.

**PCR optimisation.** We optimised our multiplex PCR using KAPA HiFi HotStart ReadyMix (Roche Diagnostics, #KK2602) formulation, which performs well for (A+T)-rich genomes like *P. falciparum*. Using mock samples of varying parasitemia, we searched for a reaction optimum in several full factorial experimental designs. For simplicity, we maintained a 2-step PCR programme and 25 µL reaction volume throughout, but explored varying 5 other factors (template DNA amount, number of cycles, primer concentration, polymerase concentration, and annealing temperature). By agarose gel, we observed that increasing the template amount increased sensitivity up to 8 µL after which there was no clear benefit; and that a programme with 35 cycles was more sensitive than 30 cycles (Supplementary Fig. 1a). A gradient PCR indicated an ideal annealing temperature of between 58–60 °C (Supplementary Fig. 1b). Increasing the amount of primer increased sensitivity but also increased visible background amplification, whereas increasing the amount of polymerase seemed to increase sensitivity without visibly affecting background (Supplementary Fig. 1c). Using this information, we selected four candidate optimal conditions, and sequenced seven mock samples under each (Supplementary Fig. 1d). From these, we selected the multiplex PCR conditions to maximise the percentage of reads mapping to *P. falciparum* malaria and mean coverage across amplicons. The final conditions were: 95 °C, 3 min; followed by 35 cycles of 98 °C 20 s, 60 °C 3 min; and a final at 60 °C for 10 min.

## NOMADS-MVP sequencing protocol

The complete laboratory protocol, including materials and primer sequences, is available online at protocols.io (English: NOMADS-MVP: Rapid Genomic Surveillance of Malaria; French: Surveillance Génomique du Paludisme par la méthode Nanopore: Protocole Rapide NOMADS-MVP). In brief, 8 µL of extracted DNA is used as template in a 25 µL multiplex PCR using 15.5 µL of Kapa HiFi HotStart ReadyMix (Roche Diagnostics, #KK2602) and 1.5 µL of the NOMADS-MVP primer pool. Multiplex PCR products are cleaned using a 0.5X ratio of AMPure XP Beads (Beckman Coulter, A63881) and eluted in 15 µL of nuclease-free water. DNA elute is quantified using the Qubit dsDNA HS Assay Kit (ThermoFisher Scientific, #Q33231) and between 200 and 800 ng of DNA is taken forward for barcoding and sequencing. We use Rapid Barcoding Kit 96 (SQK-RBK114.96) from Oxford Nanopore Technologies (ONT) following the associated ONT protocol with the following exceptions: we use 1 µL (rather than 1.5 µL) of each barcode during rapid barcoding; after barcoding we purify the pooled samples at a 0.5× ratio and elute in 15 µL; we use 800 ng of DNA for adaptor ligation, with a master mix of 1.0 µL of RA and 2.3 µL ABD (rather than 1.5 µL RA, 3.5 µL ADB). All sequencing was done using R10.4.1 flow cells on MinION Mk1B or Mk1D devices.

## Exclusion of selective whole genome amplification (sWGA)

We initially explored two versions of the protocol: one that included a selective whole genome amplification (sWGA)[50] step to preamplify bulk *P. falciparum* DNA, and one where the multiplex PCR was performed directly on extracted DNA. To this end, we developed a modified version of the selective whole-genome amplification (sWGA) protocol[50] that substituted the phi29 DNA polymerase (New England Biolabs, #M0269S) with EquiPhi29 (ThermoFisher Scientific, #A39390) as follows: 2 µLEquiPhi29 10x buffer, 0.2 µL 0.1 M DTT, 2 µL 500 µM sWGA primer pool, 1 EquiPhi29 DNA polymerase, 12.8 µL DNA; 45 °C 1 h, 60 °C 10 min; dilute before PCR with 160 µLnuclease-free water. We were able to obtain robust amplification with these conditions despite the substantial reduction in incubation time relative to phi29 DNA polymerase. We hypothesised that including sWGA might increase sensitivity, although at additional protocol cost, complexity and time. While this was confirmed with mock DBS samples created from laboratory strains, we saw no benefit in field DBS samples (Supplementary Fig. 10) and discontinued use of sWGA in August 2024.

## Real-time bioinformatics pipeline and dashboard

We developed a real-time bioinformatics pipeline and dashboard for nanopore sequencing, called *Nomadic*. The source code is publicly available on GitHub (https://github.com/JasonAHendry/nomadic) and user documentation is hosted on GitHub Pages (https://jasonahendry.github.io/nomadic/). *Nomadic* was designed for *P. falciparum* genomic surveillance with the NOMADS-MVP protocol, but was coded flexibly and supports other amplicon panels or organisms. Both the bioinformatics pipeline and dashboard are implemented in Python.

**Running a real-time analysis.** Briefly, to run *Nomadic* (v0.5.0), a user first starts a nanopore sequencing run using *MinKNOW*, ensuring to: (1) assign a unique experiment name, hereafter referred to as `<expt_name>`; (2) enable real-time basecalling using either the High Accuracy (HAC) or Super Accurate (SUP) model; (3) enable sample demultiplexing with the appropriate barcoding kit (for NOMADS-MVP, SQK-RBK114.96). The user must prepare a comma-separated values (CSV) metadata table, which maps the barcodes to sample identifiers, and save it in a *Nomadic* workspace folder with the name `<expt_name>.csv`. The user can then launch *Nomadic* from a terminal window with the command `nomadic realtime <expt_name>`.

**Implementation and bioinformatics details.** As sequencing proceeds, *MinKNOW* writes batches of basecalled reads to disk every ten minutes

as FASTQ files. These are demultiplexed by *MinKNOW* such that each barcode possesses its own folder containing associated FASTQ files. *Nomadic* keeps track of the FASTQ files in these folders and, when a new FASTQ file has been generated for a given barcode, launches a per-sample bioinformatics pipeline to process it and update results for the sample. In brief, the pipeline maps reads to the *P. falciparum* 3D7 reference genome (PlasmoDB release 67) using *Minimap2*[51] (v2.28), summarises the output of mapping (i.e., fraction of reads mapped) and coverage of the target amplicons using *samtools* (v1.17), performs variant calling with *bcftools call*[52] (v1.17) or *Delve* (v0.2.0), and annotates variants using *bcftools csq* (v1.17). Once a per-sample bioinformatics pipeline has been completed, a set of summary CSV files describing the results across all samples in the experiment is updated to reflect the incorporation of the new reads.

The dashboard runs in a separate thread and creates a graphical summary of these summary CSV files, which updates every minute. The plots are interactive, and users can hover over features of interest to open tooltips with additional information, zoom into areas of interest, or export static versions of the plots as PNG files. We implemented it using the Dash library (v2.17.1) from plotly. In our experience, sequencing and basecalling are often slower than the time *Nomadic* takes to process batches of FASTQ files, which allows the dashboard to remain up-to-date throughout the sequencing run. Once sequencing is completed, *MinKNOW* and *Nomadic* are stopped by the user. The final summary CSV files contain key results and are sufficient to reopen the dashboard, as described below.

**Viewing the dashboard for a previous experiment.** The *Nomadic* dashboard can be reopened after a sequencing experiment is completed by running the command `nomadic dashboard <expt_name>` in a terminal window.

### Sequencing coverage summary statistics

To evaluate sequencing coverage across countries and parasitemia levels, we calculated two per-sample metrics: (1) the mean coverage across all amplicons; (2) the fold-difference in coverage between the most- and least-abundant amplicon (excluding the *hrp2/3* amplicons), a measure of uniformity where a value of 1 indicates perfectly uniform coverage across amplicons. We computed these values using the `mean_cov` column in the `summary.bedcov.csv` files produced by *Nomadic*. More formally, for the per-sample fold-difference in coverage, we define $m_{ij}$ as the mean coverage in sample $i$ of amplicon $j$, and $a$ as the set of all amplicons excluding *hrp2* and *hrp3*, then we compute:

$$\frac{\max_{j \in a}(m_{ij})}{\min_{j \in a}(m_{ij})}. \tag{1}$$

Excluding *hrp2* and *hrp3* makes the statistic robust to *hrp2/3* deletions; in practice, for NOMADS-MVP, *hrp2* or *hrp3* are rarely the lowest abundance amplicon in the absence of deletion and so do not contribute to the statistic's value.

### Analysis of read lengths

As each experiment typically contains millions of reads, for computational simplicity, we included only four experiments in this analysis: from Kenya (sequenced on 2024/07/30), Mali (2024/10/25), Ethiopia (2024/11/27) and a mock sample experiment (2024/11/26). For each experiment, we randomly sampled 20 barcodes (samples) without replacement from those that: (i) had 8 or more amplicons with greater than 50× coverage; (ii) had >100 parasites/μL; (iii) did not use sWGA; (iv) were not positive or negative controls. For these barcodes, the BAM file generated by *Nomadic* was processed using *Pysam* (v0.22.1) to determine read lengths from the 'query length' field. Genomic regions of interest were delineated with a custom Python script that converted gene codon numbers of interest into genomic coordinates using the *P.*

*falciparum* 3D7 reference genome (PlasmoDB release 67) and associated general feature format (GFF) file, and then were manually verified using the Integrative Genome Viewer (IGV) (v2.5.0). We determined the percentage of reads overlapping by first using the intersect command from *bedtools*[53] (v2.31), which enabled us to filter the BAM file to only reads overlapping the region of interest, and then comparing the number of mapped reads in this BAM file against the original BAM file using *samtools* (v1.17).

### Inference of *hrp2/3* deletions

We used a set of 219 previously published samples from Asella, Ethiopia, to evaluate concordance between NOMADS-MVP and a gold-standard approach for *hrp2/3* deletion detection[34]. Recommended[54] conventional PCRs for *hrp2*[55] and *hrp3*[55] were performed in duplicate and visualised by agarose gel electrophoresis. For discordant results, we called a sample positive if there was a clear band of the appropriate length in at least one of the replicates[54]. We excluded from analysis 46 samples that failed a *kelch13* PCR, suggesting low DNA quality or parasitemia, and 24 samples that contained other *Plasmodium* species, leaving 149 samples for analysis. Of the 149 samples, 138 (92.6%) had 8 or more amplicons exceeding 50× coverage after NOMADS-MVP nanopore sequencing and were used for the comparison.

We predicted *hrp2/3* deletions from NOMADS-MVP sequencing data using a previously developed Bayesian statistical model[32]. In brief, it works as follows. For each sequencing run, the statistical model first estimates the rate of background contamination/barcode misclassification using the included negative controls. Next, it estimates the quality of each sample and average variation in quality across samples using the mean coverage over all amplicons in the multiplex PCR, excluding the *hrp2* and *hrp3* amplicons. Finally, across all samples, the posterior probability of *hrp2* and *hrp3* deletion is estimated separately using Markov Chain Monte Carlo (MCMC). Here, each chain was run for 50,000 iterations with a prior deletion probability of 0.5 for *hrp2* and *hrp3*. Although a lower prior might be justified for *hrp2*, given the prevalence of deletions in the region, in practice our estimates were insensitive to the prior. For Fig. 4a, we directly show the posterior probabilities from the model. For Fig. 4b, we are showing the maximum a posteriori (MAP) estimate of deletion. The statistical model and Bayesian MCMC were implemented in Python.

### Variant calling

We developed a novel variant caller, named *Delve*, to enable the detection of biallelic SNPs at low frequencies in polyclonal infections. *Delve* (v0.1.0) was implemented in Rust and is publicly available on GitHub (https://github.com/berndbohmeier/delve). Below, we describe the statistical model, genotype inference, bias filtering, and model tuning. The notation used throughout this section is summarised in Table 2.

**Statistical model.** *Delve* assumes genomic sites are independent and calls SNPs at one site at a time. Reads overlapping a given site are indexed by $j = 1, 2, \ldots, D$, where $D$ is the sequencing depth at the site. Each read contributes a base, $b_j$, which was aligned to the reference base REF $\in \{A, T, C, G\}$ during read mapping, and an associated Phred-scaled base quality score, $q_j$. *Delve* assumes sites are biallelic, and for each site retains only the reads carrying the reference base REF and the most common alternative base, ALT $\in \{A, T, C, G\} -$ REF; all other bases are treated as errors and discarded. Quality scores are taken from the basecaller, and then further adjusted using the Base Alignment Quality (BAQ) algorithm implemented in *samtools*[52] to account for local alignment ambiguity. Afterward, bases that have a quality score below the fixed threshold $Q_{min}$ are discarded. If the depth after filtering is less than $D_{min}$, no SNP call is made at the site.

A *P. falciparum* infection can consist of multiple clones at unknown proportions and, as a result, the fraction of clones carrying

## Table 2 | Notation for *Delve*

| Data | | |
|---|---|---|
| **Symbol** | **Description** | |
| REF | Reference base | REF $\in$ {A, T, C, G} |
| ALT | Most common alternative base | ALT $\in$ {A, T, C, G} – REF |
| $D$ | Read depth | $D \in \mathbb{Z}^+$ |
| $j$ | Read index | $j \in$ {1, 2, ..., D} |
| $b_j$ | Read base | $b_j \in$ {REF, ALT} |
| $q_j$ | Read base Phred-scaled quality score | $q_j \in \mathbb{Z}^+$ |
| $\epsilon_j$ | Read base error probability | $\epsilon_j \in$ [0, 1] |
| $v$ | Fraction of clones carrying ALT | $v \in$ [0, 1] |
| **Parameters** | | **Tuned value** |
| $D_{min}$ | Minimum depth threshold after filtering | 20 |
| $Q_{min}$ | Minimum base quality threshold | 20 |
| $v_0$ | Null-hypothesis for WSAF in LRT | 0.01 |
| $c_{LRT}$ | LRT threshold | 8 |
| $c_{ORT}$ | Strand-bias Odds Ratio Test threshold | 8 |

Symbols representing model data and parameters are separated. Data is processed one site at a time; all symbols refer to a single genomic site. *WSAF* within-sample alternative allele frequency, *LRT* likelihood-ratio test.

the alternative base, $v$, can take on any value in the interval [0, 1]. We are interested in determining $v$ from the sequencing data. Assuming that reads are independent, the likelihood for $v$ is given by:

$$\mathcal{L}(v) = \prod_{j=1}^{D} [(1-v)(1-\epsilon_j) + v\epsilon_j]^{x_j} [v(1-\epsilon_j) + (1-v)\epsilon_j]^{1-x_j}, \quad (2)$$

where $x_j = 1$ if $b_j$ = REF, and $x_j = 0$ if $b_j$ = ALT; and $\epsilon_j$ are the base error probabilities computed from the Phred-scaled quality scores, $\epsilon_j = -10\log_{10}(q_j)$.

This likelihood is a more general form of the genotype likelihood described by Li[56] (implemented in *bcftools*) as well as by McKenna et al.[57] (implemented in GATK). In those cases, the likelihood is parameterised in terms of a fixed genotype ploidy, represented by an integer. For example, in the diploid case, an organism can carry 0, 1, or 2 copies of the alternative allele, which, in the parameterisation above, corresponds exactly to $v = 0$, $v = 0.5$ or $v = 1.0$. The key difference is that in this formulation, we allow $v$ to vary continuously to reflect the unknown fraction of *P. falciparum* clones carrying the alternative base. For numerical stability, the likelihood is always evaluated in logarithmic space, yielding the log-likelihood $\ell(v)$.

**Genotype inference.** The goal of inference is to learn both $v$, the fraction of clones carrying the alternative base, and also to use this $v$ to assign a genotype for the site. First, we determine the maximum likelihood estimate of $v$, $\hat{v}_{MLE}$, using Brent's method bounded inside the interval [0, 1]. We then use the likelihood ratio test (LRT) to evaluate the evidence for a SNP at the site. In particular, we test the null hypothesis that there is no variant, $H_0$: $v \leq v_0$, versus the alternative hypothesis that there is a variant, $H_1$: $v > v_0$. In this way, to call a variant, we require that the $\hat{v}_{MLE}$ exceeds a small cut-off frequency $v_0$; in the absence of such a cut-off (i.e., if $v_0 = 0$), even the most subtle sequencing biases could eventually lead to a rejection of the null hypothesis, and spurious calling of a variant, as arbitrarily more reads were sequenced. Given these hypotheses, the LRT statistic is

$$\lambda_{LRT} = -2[\ell(\hat{v}_0) - \ell(\hat{v}_{MLE})], \quad (3)$$

where $\hat{v}_0$ is the maximum likelihood estimate (MLE) for the alternative allele frequency $v$, given the restriction $v \leq v_0$. We reject $H_0$ if $\lambda_{LRT}$ is

greater than a threshold $c_{LRT}$, which is set during model tuning. If the null hypothesis is rejected, we conduct a further likelihood ratio to distinguish between a homozygous alternative SNP, $H_0$: $v \geq (1-v_0)$, versus a heterozygous alternative SNP, $H_1$: $v < (1-v_0)$.

**SNP filtering.** *Delve* filters candidate SNPs if they exhibit excessive strand bias. In particular, we implemented the StrandOddsRatio test used in GATK[57], with minor modifications. For each candidate SNP, we count the number of reference bases on the forward ($N_{r+}$) and reverse strand ($N_{r-}$), and the number of alternative bases on the forward ($N_{a+}$) and reverse ($N_{a-}$) strand. Using these counts, we calculate the odds ratio:

$$R = \frac{N_{r+} \cdot N_{a-}}{N_{r-} \cdot N_{a+}}. \quad (4)$$

s Following GATK, we make the ratio symmetric by calculating $ORT = R + \frac{1}{R}$.

A candidate SNP is filtered from the final SNP set if *ORT* exceeds a threshold $c_{ORT}$. The rapid barcoding kit (SQK-RBK114.96) introduces strand coverage bias at amplicon edges, with the majority of coverage at the 5' and 3' ends of the amplicons deriving from the reverse and forward strands, respectively. This is an expected consequence of the transposome inserting barcodes uniformly across the amplicon and sequencing proceeding in a 3' to 5' direction. To accommodate this, we only apply the ORT filter to SNPs that have a heterozygous alternative genotype call; in this way, avoiding filtering homozygous alternative SNPs with strand coverage bias due to the barcoding chemistry.

**Model tuning.** In total, *Delve* is tuned by five parameters (Table 2). We set the base-quality filter $Q_{min}$ by examining the distribution of quality scores after applying the BAQ algorithm, observing a long tail of lower-quality bases beginning at $Q = 20$ and comprising approximately 14% of all bases. We explored varying $v_0$, $c_{LRT}$, $c_{ORT}$ to maximise recall, under the constraint of maintaining near-perfect precision in the 1000 and 10,000 parasites/µL mock DBS samples. The minimum depth filter $D_{min}$ was disabled during the downsampling analysis. Otherwise, we set $D_{min} = 20$; although in principle, the $c_{LRT}$ threshold already implicitly sets a threshold on the amount of sequencing depth (and base qualities) required to make a variant call.

## SNP calling accuracy evaluation

**Creating a set of true variants.** 3D7, Dd2, HB3 have been sequenced using Pacific Bioscience Sequencing SMRT technology and high-quality FASTA sequences are available on *PlasmoDB*[58]. To identify variants in these assemblies with respect to the 3D7 reference genome, we simulated high-quality (Phred 60) error-free reads in silico from the FASTA files, mapped them to the 3D7 reference genome with *minimap2* (v2.28), and then identified variants using the *bcftools*[52] (v1.22) `mpileup` and `call` commands. We simulated 60 error-free reads, half forward and half reverse strand, for each amplicon in NOMADS-MVP by extracting the FASTA sequence spanning ±4 kbp of the target, based on GFF files for 3D7, Dd2 and HB3. This procedure resulted in true VCF files for each of the clonal strains. We created truth VCF files for the two-strain mixtures by combining allelic depth information from the clonal strain VCF files at the correct proportions given the laboratory mixture, and updating the genotype call accordingly.

For each Cambodian strain, information about the *Kelch13* amino acid changes were available from BEI resources (IPC3445, C580Y; IPC4912, I543T; IPC5202, R539T); however, we found no information about corresponding nucleotide changes. Therefore, we performed Sanger sequencing using the NOMADS-MVP *kelch13* amplicon as a singleplex PCR to determine the *kelch13* nucleotide sequence for each strain. We used *tracy*[59] to convert the chromatogram file into a VCF with the `tracy decompose` command.

**Evaluating accuracy.** For all SNP calling accuracy evaluations, we used the Super Accurate (SUP) basecalling model in *MinKNOW* (v24.06.16) to generate FASTQ files, and then ran *Nomadic* in real-time to map reads and produce BAM files. We called variants using both the *bcftools* (v1.22) `call` command and *Delve* (v0.5.0). We used *hap.py*[60] from a *Docker* image (jmcdani20/hap.py:v0.3.12) to compute measures of variant calling accuracy in comparison to the truth VCF files described above. In particular, we used three measures of SNP calling performance: (i) the precision, $TP/(TP + FP)$, where $TP$ is the number of true positive SNP calls (homozygous or heterozygous alternative), and $FP$ is the number of false-positive SNP calls; (ii) the recall of true homozygous alternative SNPs, $TP_{homo}/(TP_{homo} + FN_{homo})$, where $TP_{homo}$ and $FN_{homo}$ are the number of true homozygous alternative SNPs called and missed, respectively; (iii) the recall of true heterozgyous alternative SNPs, $TP_{het}/(TP_{het} + FN_{het})$, which is defined analogously with (ii). We excluded from the analysis: (i) low-complexity regions of the Pf3D7 reference genome, identified using *sdust* with default parameters; (ii) the central repeat region of *csp*; (iii) the *hrp2* and *hrp3* amplicons, as their primary utility is for deletion identification and they consist largely of low-complexity repetitive sequence. This left a total of 6,681 bp for evaluation in each mock DBS sample. For amplicons where no variants existed for a particular mock sample (e.g. *Kelch13* for 3D7, Dd2 and HB3), the precision and recall are considered to be undefined; unless false-positive mutations are present, in which case the precision is zero. The within-sample alternative allele frequencies (WSAFs) plotted in Fig. 3a, c were calculated directly from the allelic depths.

**Downsampling analysis.** For the downsampling analysis, we included all mock DBS samples that: (i) were at 10,000 or 1000 parasites/μL; (ii) had greater than 500× coverage for all amplicons. This set included a total of 72 mock DBS samples (33 with 1000 parasites/μL and 39 with 10,000 parasites/μL). We excluded the mock DBS samples with 100 parasites/μL because the majority (62.2%, 28/45) did not have greater than 500× coverage for all amplicons. For each mock DBS sample, we randomly downsampled the sequencing reads to a specific mean coverage for each amplicon, using *samtools*[56] (v1.22). To achieve this, we split each mock sample's BAM file into 10 per-amplicon BAM files, each containing reads mapping to one of the NOMADS-MVP targets. We then downsampled these per-amplicon BAM files independently, with the `samtools view` command and `-s`/`-subsample` flag, to achieve a target mean coverage level, before concatenating them back into a single, per-sample BAM file; now with all amplicons having the target coverage. We downsampled to a mean coverage of 500, 400, 300, 200, 150, 100, 75, 50, and 25×, in triplicate for each mock DBS sample and coverage level, producing a total of 1944 BAM files for variant calling with *Delve* (72 samples by 9 coverage levels, in triplicate).

### Reporting summary
Further information on research design is available in the Nature Portfolio Reporting Summary linked to this article.

## Data availability
The raw sequencing data from mock DBS samples have been deposited in the NCBI Sequence Read Archive under the BioProject accession code PRJNA1401910. The raw sequencing data from field DBS samples are not publicly available due to ethical, legal, and data governance restrictions associated with the countries of origin, but may be made available by request to the corresponding author and subject to approval of the relevant national authorities. A response to requests is typically provided within 4–6 weeks. Access, if granted, will be provided for the duration and under the terms specified in the applicable data access or transfer agreements. All other data supporting the findings of the paper are available in the Source Data and Supplementary Information. Source data are provided with this paper.

## Code availability
Source code for *Nomadic* is publicly available on GitHub at: https://github.com/JasonAHendry/nomadic. Similarly, source code *Delve* is publicly available at: https://github.com/berndbohmeier/delve.

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

## Acknowledgements

The NOMADS project is funded by the Bill & Melinda Gates Foundation (INV-048316 to M.M., D.J.B. and J.A.H.). Sample collection and *hrp2/3* deletion investigation in Ethiopia was supported by the ARSUNA project,

funded by the German Federal Ministry for Economic Cooperation and Development and the Else Kröner-Fresenius Foundation (EKFS) via the Hospital Partnerships Programme (project 21Ac01060). Work in Côte d'Ivoire was funded by a grant from the Global Health Protection Programme (Federal Ministry of Health of Germany, grant number ZMII2-2523GHP029) and the WHO Hub for Pandemic Preparedness. We thank Bob Verity for helping to estimate the number of samples required to detect mutations spreading across Africa with adequate power; Olaf Kostbahn, Wendy Vienneau, and Sossena Assefa for critical financial and legal administration; and Estée Török for fruitful scientific discussions, feedback and encouragement throughout the project's duration. We are grateful to all those who participated in the studies conducted across Africa, especially the health workers and patients who made the project possible. **CDC Disclaimer:** The opinions expressed by the authors do not necessarily reflect the official views of the CDC or the U.S. Department of Health and Human Services.

## Author contributions

M.M.: Conceptualisation, methodology, validation, investigation, data curation and funding acquisition. K.M.: Methodology, validation and investigation. B.B.: Methodology, software, validation and formal analysis. M.C.: Investigation. W.V.L.: Investigation, resources and data curation. B.M.: Investigation. A.G.: Investigation. A.O.: Investigation. S.S.: Investigation. N.J.Y.: Investigation. D.S.: Investigation. B.N.: Data curation. O.Z.: Investigation. Y.N.G.T.: Investigation. F.O.: Investigation and data curation. A.Z.: Investigation. E.A.A.: Investigation. V.A.: Resources. C.C.: Investigation. S.O.: Investigation. B.O.: Investigation. M.N.: Resources. M.C.: Resources. D.K.G.: Resources. T.F.: Resources. T.B.T.: Resources. R.O.: Investigation. E.S.: Investigation. Y.S.: Resources. C.N.: Resources. O. Ouedraogo.: Resources. K.O.: Resources. O.Opaleye.: Resources. A.Olowe.: Resources. M.G.: Project administration. C.D.: Supervision. G.S.: Resources, data curation and supervision. F.P.M.: Resources and supervision. S.P.: Resources and supervision. A.D.: Resources and supervision. I.S.: Resources and supervision. D.N.: Resources and supervision. O.Ojurongbe.: Resources and supervision. S.K.: Resources and supervision. Y.P.S.: Conceptualisation, resources, data curation and supervision. J.S.S.: Resources, sata curation and supervision. M.H.: Resources and supervision. D.J.B.: Conceptualisation, data curation, supervision and funding acquisition. J.A.H.: Conceptualisation, methodology, software, validation, formal analysis, data curation, supervision, funding acquisition, and writing—original draft. All authors reviewed and approved the final version of the manuscript.

## Funding

## Competing interests

The authors declare no competing interests.

## Additional information

**Mulenga Mwenda**[1,17], **Karolina Mosler** [2,17], **Bernd Bohmeier** [2], **Miriam Chomba**[3], **Welmoed van Loon** [4], **Brenda Mambwe**[1], **Amy Gaye**[5], **Adedolapo Olorunfemi**[6], **Salma Suliman**[6], **Nassandba Julien Yanogo**[7], **Djiby Sow** [5], **Bassirou Ngom**[5], **Oumou Aïcha Zeïna Zoure**[7], **Yssimini Nadège Guillène Tibiri**[7], **Fiyinfoluwa Ojeniyi**[6], **Arsène Zongo** [8], **Etilé A. Anoh**[9], **Vincent Achi**[9], **Carol Chiyesu**[1], **Sheila Otieno**[3], **Bixa Ogola** [3], **Moussa Niangaly**[2,10], **Manuela Carrasquilla**[2], **Dagaga Kenea Goboto**[11,12], **Torsten Feldt**[11,13], **Tafese Beyene Tufa**[11,12,13], **Rafael Oliveira**[4], **Emma Schallenberg**[4], **Yuhana Sogoba**[10], **Christina Ntalla**[2], **Oumarou Ouedraogo**[7], **Kephas Otieno**[3], **Oluyinka Opaleye**[6,14], **Adekunle Olowe**[6,14], **Marley Gibbons**[1], **Chris Drakeley**[1], **Grit Schubert**[8], **Frank P. Mockenhaupt**[4], **Silvia Portugal** [2], **Awa B. Deme**[5], **Issiaka Soulama** [7], **Daouda Ndiaye**[5], **Olusola Ojurongbe**[6,14], **Simon Kariuki**[3], **Ya Ping Shi**[15], **Jonathan S. Schultz** [15], **Moonga Hawela**[16], **Daniel J. Bridges** [1] & **Jason A. Hendry** [2] ✉

[1]PATH, Lusaka, Zambia. [2]Max Planck Institute for Infection Biology, Berlin, Germany. [3]Centre for Global Health Research, Kenya Medical Research Institute, Kisumu, Kenya. [4]Charité – Universitaetsmedizin Berlin, Charité Center for Global Health, Institute of International Health, Berlin, Germany. [5]Centre International de recherche, de Formation en Genomique Appliquee et de Surveillance Sanitaire (CIGASS), Dakar, Senegal. [6]Center for Emerging and Re-emerging Infectious Diseases, Ladoke Akintola University of Technology, Ogbomoso, Nigeria. [7]Department of Biomedical and Public Health, Health Sciences Research Institute (IRSS), National Center for Scientific and Technological Research (CNRST), Ouagadougou, Burkina Faso. [8]Unit for Public Health Laboratory Support, Centre for International Health Protection (ZIG), Robert Koch Institute, Berlin, Germany. [9]Centre Hospitalier et Universitaire de Bouaké, Bouaké, Côte d'Ivoire.

[10]Parasites and Microbes Research and Training Center (MRTC), University of Science, Techniques and Technologies (USTTB), Bamako, Mali. [11]Hirsch Institute of Tropical Medicine, Asella, Ethiopia. [12]College of Health Sciences, Arsi University, Asella, Ethiopia. [13]Department of Gastroenterology, Hepatology and Infectious Diseases, University Hospital Düsseldorf, Heinrich Heine University, Düsseldorf, Germany. [14]Department of Medical Microbiology and Parasitology, Ladoke Akintola University of Technology, Ogbomoso, Nigeria. [15]Division of Parasitic Diseases and Malaria, National Center for Emerging and Zoonotic Diseases, Centers for Disease Control and Prevention, Atlanta, GA, USA. [16]National Malaria Elimination Centre, Chainama, Lusaka, Zambia. [17]These authors contributed equally: Mulenga Mwenda, Karolina Mosler. ✉e-mail: hendry@mpiib-berlin.mpg.de

