## [Peer Review file · Nature Communications]

Continental-scale genomic surveillance of *Plasmodium falciparum* malaria across sub-Saharan Africa with rapid nanopore sequencing

Corresponding Author: Dr Jason Hendry

Version 0:

Reviewer comments:

Reviewer #1

(Remarks to the Author)

'Continental-scale genomic surveillance of *Plasmodium falciparum* malaria with rapid nanopore sequencing', Mwenda et al.

Overall impression:

This is a great piece of work: commendable in scale, ambition, collaborative approach, and clarity of write-up and figures. It will be of high interest to the malaria genomics and nanopore research communities. I recommend it strongly for publication.

Major comments:

(I called these 'major' but would still recommend for publication regardless).

There isn't much reporting of population genetic results in the main paper, eg drug resistance marker frequencies, *csp* variation, *ama1* etc. Surely this is worth commenting upon/ a figure? You could compare the allele frequencies to known MalariaGEN allele frequencies as a sanity check.

I personally am always cautious with *hrp2/3* about calling deletions, when the test relies on a negative result ie drop in coverage. This is especially complicated with polyclonal infections in which some parasites have a deletion and some don't. And the *hrp2/3* breakpoints are really complicated. There is already a WHO approved *hrp2/3* RT-PCR assay. It may be worth exploring this a bit further in the Discussion, ie 1) Proving a negative is inherently problematic, 2) a combined approach is probably needed eg RDT pattern (LDH+ but *hrp*-) plus molecular evidence, 3) mention there is already a WHO approved approach. Personally I am agnostic whether we need a sequencing-based approach to *hrp2/3* when we can use existing tools (microscopy, RDT, and PCR).

Minor comments:

"Perhaps 50,000 to 100,000 samples would need to be sequenced annually to have adequate statistical power to detect emerging mutations across the entire region"

- I don't disagree but I'm curious to know if there is a good citation for this number, how was it reached?

"We examined the effect of parasitemia on sequencing coverage"

- It is very common in endemic areas to measure parasitaemia as parasites per 200 white blood cells. I wonder if the authors could add an example cutoff in parasites per 200 WBC, given typical assumptions of RBC and WBC counts, at least for a rough guide that would be directly relevant to microscopists working in the field?

I agree with the authors for including the section on parasitaemia in the Discussion ie their assay may not be optimal for sub-microscopic parasitaemias. However (and to the authors' defence), my own view is that people can get too focused on very low parasitaemia sampling in the context of surveillance. I agree with the authors' introductory remark that there should be many thousands of parasites being sampled across endemic parts of Africa each year for statistical power of malaria surveillance, and we are way below that at current. The nanopore-based assays described here and by other groups perform well at most clinical parasitaemias. Artemisinin partial resistance, partner drug resistance, & diagnostic test resistance are all on the rise in parts of Africa and the *csp* vaccines are being rolled out. The situation is urgent! I am a

proponent of the view, 'don't let the best be the enemy of the good'. These assays, using samples from typical clinical parasitaemias, are needed now and I hope they are scaled up in Africa ASAP. Pushing into sub-microscopic parasitaemias is of academic interest but should not delay the urgently needed expansion of malaria molecular surveillance tools in Africa, and clinical-range parasitaemia sampling is adequate for this, providing a reasonable parasitaemia cut-off is used to prevent wastage. I wonder, where would the authors place this cutoff pragmatically (in parasites/ul, parasites/ 200 WBC, and % infected RBCs units)?

In the Discussion - personally I would further emphasise that we now really need these molecular surveillance tools to go from research to clinical & public health diagnostics. That needs a switch in focus from researcher-based R&D to industry & regulatory bodies going through the necessary quality assurance and licensing steps for in vitro diagnostic devices. I think the tools are there - it needs the buy-in from industry and public health now.

I commend the authors on having their protocols available in both English and French.

I commend the authors for developing a bespoke genotyping method that accounts for polyclonal infections, Delve. We developed a method for testing performance of mixed sample haplotype calls: <https://verixiv.org/articles/2-146>, GitHub links at the bottom of the page. The workflow takes reads from pure clones and mixes them to create read mixtures with known ground truth at any potential ratio. The authors could use a similar approach to test Delve (just a suggestion/ idea, not essential). We also have some laboratory clone mixtures available if the authors would like to test Delve using these too - the genome data is available in the GitHub repos linked to in that paper. Again, not essential, just an idea. It would be good to have more benchmarking of malaria parasite genotyping/ haplotype calling methods but I recognise that lies beyond the scope of this manuscript. I do think it's a shame to lose the haplotypes from long-read amplicons, as that is a key benefit of ONT compared to Illumina, so perhaps that could be further highlighted in the Discussion eg are there scenarios where haplotype knowledge is important for surveillance eg perhaps with csp?

Overall this is a great piece of work and I'll be glad to see it published. I hope this 'makes a splash' sufficient to get industry interested so this can be rolled out as a diagnostic assay run by clinical and public health labs across endemic countries. The time is now!

- Dr William L. Hamilton, Cambridge University Department of Medicine & Cambridge University Hospitals NHS Foundation Trust.

(Remarks on code availability)

Reviewer #2

(Remarks to the Author)

In the evolving era of translational genomic surveillance, this paper by the NOMAD consortium is timely for the malaria elimination research community in Africa. It adds to the tools available to expand molecular surveillance of malaria parasites, especially in resource-limited settings. Beyond the amplification and sequencing, the team also developed analysis tools that simplify the process from sample to data without the need for expert bioinformaticians. Overall, they have clearly demonstrated the following

1. NOMAD-MVP is cheap and easy to implement
2. Results are comparable across multiple settings
3. Both drug resistance and diagnostic markers can be genotyped
4. Results are valid against known mock reference samples

The successful implementation of the approach and protocol in both trained and first-time-use labs across Africa is a testament to the utility and ease of transfer of the protocol across the continent. However, most of the participating institutions in the current paper are, on average, more advanced than most public health facilities in Africa. Future exploration of the transfer to a public health lab supporting malaria elimination countries will increase translation.

Although some asymptomatic samples were included, the sensitivity is at best at high parasite loads. Like all other surveillance needed for malaria, accessing and evaluating the large number of asymptomatics, likely with lower parasitemia, will enrich the uptake and translation of data. This is particularly important for areas with decreasing transmission moving towards pre-elimination. As the lower parasitemias in asymptomatics will be challenging, some of the data from SWGA applied to lower parasitemia samples, reflective of what most pre-elimination settings, could have been included, given SWGA was already tested.

The current tool targets legacy antimalarial resistance markers and emerging threats from HRP2 and Kelch13. As new targets for other drugs are emerging, the authors could include, in perspective, the plan and ease of adding new targets to the panel. Resistance to an important partner drug, lumefantrine, is now associated with the new locus PX1, which would immediately be beneficial for translation by NMCPs.

In the methods, except for validation of the HRP2/3 deletion, the mock samples used were lab strains with known genotypes. For Kelch13, these included Cambodian isolates with known variants. Known reference isolates with characterised K13 true

variants relevant to emerging partial ART resistance in East Africa could have added to the validation.

Ama1 and csp were sequenced, and these can be used to determine the complexity of infection, an index that is broadly informative of the intensity of transmission. Although the title is indicative of genomic surveillance, leaving this index from the current paper is disappointing. Mindful of the authors indicating that this is in plans, the methods of variant calling by Delve have been refined for biallelic alternative variants, representing as low as 5% in mixed infections. Will this be the case for the more diverse ama1??

Overall, this is an important development and the new NOMAD-MVP is an excellent additional tool to transform speed of acquisition and use of MMS data in Africa.

(Remarks on code availability)

Reviewer #3

(Remarks to the Author)

This study addresses critical technical bottlenecks in the genomic surveillance of *Plasmodium falciparum* malaria in sub-Saharan Africa by developing a low-cost, rapid nanopore sequencing protocol—complemented by a locally deployable bioinformatics dashboard (Nomadic) and a novel variant-calling tool (Delve) capable of detecting low-frequency clonal variants. The authors further demonstrate the feasibility of decentralized implementation through the successful local sequencing of 1,065 clinical samples across six African countries, underscoring both technical innovation and public health relevance. Nevertheless, several aspects of the study require further clarification and strengthening.

1. It is recommended to further analyze the causes of differences in sequencing qualification rates (e.g., 95.1%–96.9% in Kenya vs. 62.1%–69.0% in Côte d'Ivoire) and coverage uniformity (fold-difference = 44.4× in Zambia) among samples from different countries. Additionally, the impact of differences in storage, transportation, and operational procedures on results between samples sequenced externally (Ethiopia/Mali) and those sequenced locally (the other 6 countries) should be evaluated.

2. Delve was only compared with bcftools. It is suggested to add comparisons with other tools applicable for detecting low-frequency clones of *Plasmodium* (e.g., SeekDeep) and conduct parameter sensitivity analysis.

3. To enhance the reproducibility of the study, it is recommended to upload raw sequencing data to public databases or other accessible platforms.

4. It is advisable to further analyze whether there are differences in detection accuracy and sensitivity between nanopore sequencing and mainstream methods such as Illumina sequencing.

5. It's suggested to clarify the typical *Plasmodium* density in patients under normal circumstances, explain how the threshold of 100 parasites/μL (mentioned in the manuscript) is defined, and note that this threshold seems relatively high.

(Remarks on code availability)

Version 1:

Reviewer comments:

Reviewer #1

(Remarks to the Author)

The authors have addressed all of my comments. This is a high quality piece of work and I recommend the article is published.

(Remarks on code availability)

Reviewer #2

(Remarks to the Author)

The authors have responded in detail to all the concerns raised. I am satisfied with the responses. The figures showing the superior performance of DELVE vs LoFreq are appreciated.

(Remarks on code availability)

Reviewer #3

(Remarks to the Author)

The authors have addressed my concerns, and I have no further issues with the manuscript in its current form.

(Remarks on code availability)

REVIEWER COMMENTS

Reviewer #1 (Remarks to the Author):

'Continental-scale genomic surveillance of *Plasmodium falciparum* malaria with rapid nanopore sequencing',
Mwenda et al.

Overall impression:

This is a great piece of work: commendable in scale, ambition, collaborative approach, and clarity of write-up and figures. It will be of high interest to the malaria genomics and nanopore research communities. I recommend it strongly for publication.

Thank you for such positive and encouraging feedback on our work.

Major comments:

(I called these 'major' but would still recommend for publication regardless).

There isn't much reporting of population genetic results in the main paper, eg drug resistance marker frequencies, *csp* variation, *ama1* etc. Surely this is worth commenting upon/ a figure? You could compare the allele frequencies to known MalariaGEN allele frequencies as a sanity check.

It is true that we do not report population genetic findings in the paper. We made the decision not to include these results for two main reasons: (i) to keep the paper focussed on technical validation of the novel methods we developed (e.g. NOMADS-MVP, *Nomadic, Delve*), rather than epidemiological findings; and (ii) to allow our co-authors to lead in publishing their own country's population genetic data, which they all are preparing to do in the near future.

With respect to our aim of SNP calling validation, we agree that comparing population-level allele frequencies to those generated by MalariaGEN can be a compelling analysis (e.g. as your team did with *csp* in Ghana). However, it is indeed more of a sanity check than a formal validation; concordance can depend on having sequenced samples from a closely matched geography / time as MalariaGEN, and any observed differences could be due to e.g. population structure and/or sampling, rather than indicating any technical biases with nanopore. Instead in Fig 3, we formally evaluate SNP calling performance across a variety of drug resistance markers, as well as the SNPs in *ama1* and the C-terminal domain of *csp*. For now we hope it is adequate to leave the population genetic findings to future papers, which can focus in more detail on the findings from each country and their epidemiological relevance.

I personally am always cautious with *hrp2/3* about calling deletions, when the test relies on a negative result ie drop in coverage. This is especially complicated with polyclonal infections in which some parasites have a deletion and some don't. And the *hrp2/3* breakpoints are really complicated. There is already a WHO approved *hrp2/3* RT-PCR assay. It may be worth exploring this a bit further in the Discussion, ie 1) Proving a negative is inherently problematic, 2) a combined approach is probably needed eg RDT pattern (LDH+ but *hrp*-) plus molecular evidence, 3) mention there is already a WHO approved approach. Personally I am agnostic whether we need a sequencing-based approach to *hrp2/3* when we can use existing tools (microscopy, RDT, and PCR).

These are all valid points about the difficulties of *hrp2/3* deletion detection and we have added a paragraph to the discussion highlighting them. While we acknowledge that targeted sequencing is not (and probably should not) be the only approach for molecular confirmation of *hrp2/3* deletions, we do feel it has some advantages. In the WHO's master protocol for *hrp2/3* deletion surveillance (<https://www.who.int/publications/i/item/9789240099951>), four methods are described (pg. 16–17, Table 5): conventional PCR, RT-qPCR, ddPCR, and sequencing. Compared to a conventional PCR workflow (e.g. <https://doi.org/10.1186/1475-2875-13-283> or <https://doi.org/10.1186/s12936-018-2287-4>), NOMADS-MVP is more streamlined as it simultaneously assays both *hrp2* and *hrp3* and has multiple internal single-copy gene controls (e.g. all the other non-*hrp* targets). Compared to RT-qPCR or ddPCR, the equipment costs of NOMADS-MVP are lower (e.g. a MiniON and laptop vs a multichannel qPCR machine); and anecdotally we've heard it can be tricky to get ddPCR/qPCR set up reliably in new labs, whereas setting up nanopore sequencing is relatively straightforward.

Another perspective is that there are cases where it can be advantageous to have *bundled* diagnostic and drug resistance surveillance, which sequencing uniquely enables. One is in regions where both drug- and diagnostic resistance are an active concern and would otherwise be surveyed separately (e.g. Ethiopia, Eritrea); here

getting both outputs from a single assay could save time/cost, simplify procurement, and potentially boost power (depending on the study design). Another potentially useful context is in regions where the risk of *hrp2/3* deletion is quite low (e.g. those identified as low-risk here: <https://www.nature.com/articles/s41591-025-03974-3>). While it is still important to conduct *hrp2/3* deletion surveillance in these regions, running stand-alone studies for *hrp2/3* can be a lot of effort / time in sample collection to address a relatively small risk. If sequencing is used, the effort put into study design and sample collection can provide information about drug resistance markers, as well as on potential *hrp2/3* deletions.

Minor comments:

"Perhaps 50,000 to 100,000 samples would need to be sequenced annually to have adequate statistical power to detect emerging mutations across the entire region"

- I don't disagree but I'm curious to know if there is a good citation for this number, how was it reached?

We made a back-of-the-envelope estimate of how many samples would be required to reliably detect if an emerging mutation exceeded ~1-2% prevalence anywhere across malaria-endemic Africa, as follows:

Let our goal be to detect, with at least 80% power, a mutation with prevalence p at a given sampling location. For example, the mutation could be a novel *kelch13* variant and the sampling location could be a health facility. Assuming random independent sampling, the probability of detecting the mutation in at least one sample, which is equivalent to the power, is $1 - (1 - p)^n$, where n is the number of samples sequenced. With some algebraic rearrangement, we can use this formula to calculate that for 80% power: $n = 161$ when $p = 0.01$; or $n = 80$ when $p = 0.02$.

There are 44 malaria-endemic countries in Africa and we posit (rather arbitrarily) that 15 collection sites per country is reasonable (some countries e.g. Nigeria will require more; and some e.g. Burundi less). Then we would need to sequence 106,260 ($= 44 \times 15 \times 161$) to 52,800 ($= 44 \times 15 \times 80$) samples to detect emerging mutations at 1 to 2% prevalence, respectively.

We now include this calculation in Supplementary Note 1.

"We examined the effect of parasitemia on sequencing coverage"

- It is very common in endemic areas to measure parasitaemia as parasites per 200 white blood cells. I wonder if the authors could add an example cutoff in parasites per 200 WBC, given typical assumptions of RBC and WBC counts, at least for a rough guide that would be directly relevant to microscopists working in the field?

In the Results section, we have now defined the parasitemia thresholds in terms of parasites per WBC and iRBC %.

I agree with the authors for including the section on parasitaemia in the Discussion ie their assay may not be optimal for sub-microscopic parasitaemias. However (and to the authors' defence), my own view is that people can get too focused on very low parasitaemia sampling in the context of surveillance. I agree with the authors' introductory remark that there should be many thousands of parasites being sampled across endemic parts of Africa each year for statistical power of malaria surveillance, and we are way below that at current. The nanopore-based assays described here and by other groups perform well at most clinical parasitaemias. Artemisinin partial resistance, partner drug resistance, & diagnostic test resistance are all on the rise in parts of Africa and the csp vaccines are being rolled out. The situation is urgent! I am a proponent of the view, 'don't let the best be the enemy of the good'. These assays, using samples from typical clinical parasitaemias, are needed now and I hope they are scaled up in Africa ASAP. Pushing into sub-microscopic parasitaemias is of academic interest but should not delay the urgently needed expansion of malaria molecular surveillance tools in Africa, and clinical-range parasitaemia sampling is adequate for this, providing a reasonable parasitaemia cut-off is used to prevent wastage. I wonder, where would the authors place this cutoff pragmatically (in parasites/ul, parasites/200 WBC, and % infected RBCs units)?

We are strongly aligned with you on this point; we feel that for widespread and routine MMS, NGS assays with submicroscopic sensitivities are not necessary and, in fact, can be disadvantageous -- the higher the sensitivity of an assay, the more sensitive it becomes to contamination, often making it more technically difficult to deploy. For applications where the study is focussed on clinical cases, the additional risk of contamination is taken on with little benefit. Unfortunately, there does seem to be a tendency in the community to largely judge an assay by its limit-of-detection (LoD) with respect to parasitemia; perhaps because the LoD is very commonly reported/measured, whereas other features of practical importance for deploying an assay (e.g. setup costs, protocol and analysis complexity) are less routinely and rigorously reported.

We have updated our reporting of our parasitemia cut-off in the Results section '*Robust sequencing coverage across countries and parasitemia levels*' terms of more pragmatic measures, as you suggested.

In the Discussion - personally I would further emphasise that we now really need these molecular surveillance tools to go from research to clinical & public health diagnostics. That needs a switch in focus from researcher-based R&D to industry & regulatory bodies going through the necessary quality assurance and licensing steps for in vitro diagnostic devices. I think the tools are there - it needs the buy-in from industry and public health now.

We agree and have now emphasised the need for this translation in the last paragraph of the discussion.

I commend the authors on having their protocols available in both English and French.

Thank you, our french-speaking co-authors from West Africa led this effort.

I commend the authors for developing a bespoke genotyping method that accounts for polyclonal infections, Delve. We developed a method for testing performance of mixed sample haplotype calls: <https://verixiv.org/articles/2-146>, GitHub links at the bottom of the page. The workflow takes reads from pure clones and mixes them to create read mixtures with known ground truth at any potential ratio. The authors could use a similar approach to test Delve (just a suggestion/ idea, not essential). We also have some laboratory clone mixtures available if the authors would like to test Delve using these too - the genome data is available in the GitHub repos linked to in that paper. Again, not essential, just an idea. It would be good to have more benchmarking of malaria parasite genotyping/ haplotype calling methods but I recognise that lies beyond the scope of this manuscript. I do think it's a shame to lose the haplotypes from long-read amplicons, as that is a key benefit of ONT compared to Illumina, so perhaps that could be further highlighted in the Discussion eg are there scenarios where haplotype knowledge is important for surveillance eg perhaps with csp?

Thank you for highlighting the availability of both your *in silico* tools and laboratory mixtures to further benchmark *Delve*. We agree that losing haplotype information is not ideal, especially given that it is a strength of the ONT platform. Our plan for future work is to continue to extend *Delve* to enable haplotype inference (e.g. with phasing after SNPs are called) or potentially building a new assembly / haplotyping tool to handle the fragmented rapid kit reads, if no existing tools proves adequate. We would be very interested in exploring using your *in silico* tools / lab mixtures strains once we undertake that work, but feel it is outside the scope of the current manuscript.

We have now highlighted the value of haplotypes in relation to *csp* surveillance in the Discussion.

Overall this is a great piece of work and I'll be glad to see it published. I hope this 'makes a splash' sufficient to get industry interested so this can be rolled out as a diagnostic assay run by clinical and public health labs across endemic countries. The time is now!

Again, thank you for your thoughtful and encouraging review.

- Dr William L. Hamilton, Cambridge University Department of Medicine & Cambridge University Hospitals NHS Foundation Trust

Reviewer #2 (Remarks to the Author):

In the evolving era of translational genomic surveillance, this paper by the NOMAD consortium is timely for the malaria elimination research community in Africa. It adds to the tools available to expand molecular surveillance of malaria parasites, especially in resource-limited settings. Beyond the amplification and sequencing, the team also developed analysis tools that simplify the process from sample to data without the need for expert bioinformaticians. Overall, they have clearly demonstrated the following

1. NOMAD-MVP is cheap and easy to implement
2. Results are comparable across multiple settings
3. Both drug resistance and diagnostic markers can be genotyped
4. Results are valid against known mock reference samples

Thank you for your positive and thorough feedback on our work. We hope the features of NOMADS-MVP you describe will make it a useful addition to the methods supporting *P.f.* genomic surveillance across malaria-endemic Africa.

The successful implementation of the approach and protocol in both trained and first-time-use labs across Africa is a testament to the utility and ease of transfer of the protocol across the continent. However, most of the participating institutions in the current paper are, on average, more advanced than most public health facilities in Africa. Future exploration of the transfer to a public health lab supporting malaria elimination countries will increase translation.

We agree that the prior NGS experience (and/or expertise in *P. f.* malaria molecular analyses) of our collaborating institutions exceeds that of average public health facilities. We also agree this is a relevant fact for readers, and in the Results section 'Implementation at scale across sub-Saharan Africa', we have outlined the experience / infrastructure of the participating institutions. In addition, we now highlight at the end of the Discussion the need to focus on future translation to public health laboratories.

Although some asymptomatic samples were included, the sensitivity is at best at high parasite loads. Like all other surveillance needed for malaria, accessing and evaluating the large number of asymptomatics, likely with lower parasitemia, will enrich the uptake and translation of data. This is particularly important for areas with decreasing transmission moving towards pre-elimination. As the lower parasitemias in asymptomatics will be challenging, some of the data from SWGA applied to lower parasitemia samples, reflective of what most pre-elimination settings, could have been included, given SWGA was already tested.

Thank you for your suggestion. We found that sWGA did not improve the performance of our assay on low parasitemia samples and hence we stopped using it early on in the project, as it added costs and complexity to the protocol. This is described in the Methods section 'Exclusion of selective whole genome amplification (sWGA)' and we have now added a supplementary figure (SFig. 10) documenting these results. Moreover, in the Discussion, we highlight that generating more sensitive panels would likely require using smaller amplicons; but this also brings disadvantages, as it means more primers and PCRs are required to cover the same target regions. Regardless, we agree that in the future it may be necessary to develop more sensitivity assays that are better suited for pre-elimination contexts. However at the present, we are confident the NOMADS-MVP is sufficiently sensitive for many important applications.

The current tool targets legacy antimalarial resistance markers and emerging threats from HRP2 and Kelch13. As new targets for other drugs are emerging, the authors could include, in perspective, the plan and ease of adding new targets to the panel. Resistance to an important partner drug, lumefantrine, is now associated with the new locus *Px1*, which would immediately be beneficial for translation by NMCPs.

This is a great point and something we are actively working on. In our previous publication we created open-source software for multiplex PCR primer design called *multiply* (<https://www.nature.com/articles/s41467-024-45688-z>). We agree that *Px1* is a compelling new marker for lumefantrine resistance, and we hope to incorporate it (as well as other interesting markers, e.g. *PfCoronin*) into NOMADS panels as soon as possible.

In the methods, except for validation of the HRP2/3 deletion, the mock samples used were lab strains with known genotypes. For Kelch13, these included Cambodian isolates with known variants. Known reference isolates with characterised K13 true variants relevant to emerging partial ART resistance in East Africa could have added to the validation.

We would have liked to add such an analysis, but did not have access to samples carrying the new *kelch13* mutations (e.g. C469Y, A675V, R561H) to allow it. We hope in the future, as we grow collaborations in East Africa, we will be able to conduct exactly these sorts of studies.

Ama1 and *csp* were sequenced, and these can be used to determine the complexity of infection, an index that is broadly informative of the intensity of transmission. Although the title is indicative of genomic surveillance, leaving this index from the current paper is disappointing. Mindful of the authors indicating that this is in plans, the methods of variant calling by *Delve* have been refined for biallelic alternative variants, representing as low as 5% in mixed infections. Will this be the case for the more diverse *ama1*??

We agree and hope soon to incorporate a robust COI estimation metric into the NOMADS bioinformatic pipeline. In this manuscript, we focussed first on drug resistance and diagnostic test failure, as they are more urgent for most public health applications of genomics than transmission intensity estimation. Moreover, as you suggest, most empirical work to-date has found that COI is really only broadly indicative of transmission; in many cases it has been quite a noisy relationship (e.g. <https://www.nature.com/articles/s41467-020-15779-8>).

For *ama1*, at COI = 2 the minor clone LoD is 5% as described in the manuscript. For COI > 2, the higher diversity of *ama1* would indeed result in the biallelic assumption in our model being violated at more sites than e.g. across drug resistance genes. However, if during COI estimator validation this was problematic, we could extend *Delve* to a multiallelic model to accommodate (the mathematical changes are straightforward, but the implementation would require more work and parameter tuning). For now, however, we felt we did not want to further delay the publication of this work to develop a COI estimator, as many key aspects for malaria control (drug resistance, *hrp2/3* deletions) were already validated and could be helpful immediately.

Overall, this is an important development and the new NOMAD-MVP is an excellent additional tool to transform speed of acquisition and use of MMS data in Africa.

Thank you again for your time and thoughtful feedback our work.

Reviewer #3 (Remarks to the Author):

This study addresses critical technical bottlenecks in the genomic surveillance of *Plasmodium falciparum* malaria in sub-Saharan Africa by developing a low-cost, rapid nanopore sequencing protocol—complemented by a locally deployable bioinformatics dashboard (Nomadic) and a novel variant-calling tool (Delve) capable of detecting low-frequency clonal variants. The authors further demonstrate the feasibility of decentralized implementation through the successful local sequencing of 1,065 clinical samples across six African countries, underscoring both technical innovation and public health relevance. Nevertheless, several aspects of the study require further clarification and strengthening.

Thank you for your time and the thoughtful review of our manuscript.

1. It is recommended to further analyze the causes of differences in sequencing qualification rates (e.g., 95.1%–96.9% in Kenya vs. 62.1%–69.0% in Côte d'Ivoire) and coverage uniformity (fold-difference = 44.4x in Zambia) among samples from different countries. Additionally, the impact of differences in storage, transportation, and operational procedures on results between samples sequenced externally (Ethiopia/Mali) and those sequenced locally (the other 6 countries) should be evaluated.

We have conducted a new analysis looking at sequencing coverage and pass rates as a function of clinical vs asymptomatic samples, and international vs local sequencing (Supplementary Fig. 5) In both cases, although these factors do have some effect on sequencing pass rate (e.g. asymptomatic pass rate 58–83%; clinical 67–90%), the parasitemia of a sample has a stronger (and probably more direct) influence. With regards to the poorer coverage uniformity in Zambia, this was due to lower coverage of *dhps/dhfr* and we now mention this in the manuscript (the cause is not completely certain, but it may be due to the primer pool that was used).

Beyond this, unfortunately, it is hard to pinpoint systematic factors that explain variation between countries using our data (e.g. such as storage, transportation, etc.). This is primarily because the number of potentially relevant factors that vary between each country is very large, including study designs and sample sets, laboratory environments and equipment, experience with molecular methods and NGS, reagent batches, etc.. In a majority of cases, it is very difficult to know which of these factors most influenced sequencing performance in a country; most-likely it is a combination of many factors. In other cases, we do have some specific probable causes for failure, but they are not particularly generalisable / relevant in the context of the manuscript. For example, in Burkina Faso, we initially ordered the wrong PCR plates for the available thermalcycler, and this led to some evaporation of samples which manifested as QC failures later on. Regarding the lower pass rate in Côte d'Ivoire, the team there was the only one which undertook sequencing without any prior training as part of a NOMADS workshop, and in retrospect we believe that some DNA may have been lost during the AMPure XP DNA clean-up step, which is one of the most difficult hands-on steps in the protocol (this would explain why some high parasitemia samples failed; however, we can't be completely certain). For these issues, we have made additional comments in our publicly available protocol, which we feel is the most appropriate place for them, rather than in the main manuscript. In general, we feel presenting all the data in this way is valuable as it sets a very realistic expectation of the variation in performance of our assay across a multitude of laboratories; although, as your comment alludes, a disadvantage is that some of the causes of this variation will be difficult to completely explain. In the future, a deeper understanding how specific factors like storage material or sample transportation / age influence sequencing performance could be determined with experimental designs in which these factors are varied in a more controlled manner.

2. Delve was only compared with bcftools. It is suggested to add comparisons with other tools applicable for detecting low-frequency clones of *Plasmodium* (e.g., SeekDeep) and conduct parameter sensitivity analysis.

We have now conducted a parameter sensitivity analysis for *Delve* exploring the effect of varying the likelihood-ratio test threshold ($CLRT$) and odds-ratio bias filter ($CORT$) on SNP calling precision and recall across all 135 mock samples we evaluated (Supplementary Fig. 6). The likelihood-ratio test threshold ($CLRT$), in particular, controls the stringency of SNP calling in *Delve* and was set with a default value of 8 in the manuscript. As expected, the parameter sensitivity analysis revealed that reducing $CLRT$ below the default value causes a loss of precision (i.e. an increase in false-positive SNP calls) and is not recommended. Increasing the $CLRT$ to a value of 50 improved precision for 100p/uL samples (94.6%, $CLRT = 8$; 98.9%, $CLRT = 50$) while, interestingly, still enabling high recall of SNPs carried by clones down to 10% across all parasitemias (compared to 5% detection with the more permissive threshold). This analysis suggests that for applications where a high precision in low parasitemia samples of high importance, boosting the $CLRT$ may be advisable.

Furthermore, we have conducted a comparison of *Delve* against a somatic variant caller named *LoFreq* (<https://pubmed.ncbi.nlm.nih.gov/23066108/>), which does not make the diploid human assumption that is made by *bctools* and other variants callers designed for nanopore data (e.g. *Clair3*, *DeepVariant*). In comparative analyses on other datasets, *LoFreq* performs well against other viral/bacterial variant callers on Illumina data (e.g. <https://academic.oup.com/bib/article/22/3/bbaa123/5868070>). Unfortunately however, it does not actually provide genotype calls -- it only indicates variant positions. As a result, we had to develop an additional algorithm and code to post-process *LoFreq* outputs into genotype calls. In the end, *LoFreq* performed worse than *Delve* both with the full data and in downsampled replicates (Fig. 1 and 2, below). Moreover, on the mock DBS samples in it ran on average 12x slower than *Delve* (mean 5.6s *Delve*; mean 69.5s *LoFreq*). Because of the need for additional postprocessing of *LoFreq* files to even conduct this comparison, we felt it was not particularly suitable or relevant for incorporation into the main manuscript.

We agree that comparing to *SeekDeep* (or *DADA2*) would be most relevant, but unfortunately these bioinformatic tools do not work for our data and so we are unable to do so. This is because both tools require input reads to cover *complete* amplicons (which are used for clustering); and our reads are cut into pieces by the rapid barcoding kit. As a result, the core clustering algorithms inside of *DADA2* and *SeekDeep* do not work on our FASTQ files. This was a major reason that we felt it important to develop a new bioinformatic method in the first place. In the future, we hope to find ways to use *SeekDeep* on our data (we are in conversation with developer Nick Hathaway about how this might be possible) or, if necessary, develop a novel haplotyping tool that can handle read fragmentation. Please note that we have highlighted these issues in the Paragraph 3 of the discussion.

Figure 1. Comparison between *Delve* and *LoFreq*. Recall of heterozygous ALT (Recall_het), homozygous ALT (Recall_homalt), and precision (Precision_al) are shown for different mock samples and mixture proportions (x-axis). Notice that *Delve* (red) outperforms *LoFreq* (blue) in nearly all cases.

Figure 2. Comparison between Delve and LoFreq with downsampled data. Same as Figure 1, but data has been randomly downsampled in triplicate for each mock sample. Again Delve (red) outperforms LoFreq (blue) in nearly all cases.

3. To enhance the reproducibility of the study, it is recommended to upload raw sequencing data to public databases or other accessible platforms.

We have uploaded the raw sequencing data for all mock DBS samples sequenced at the Max Planck Institute of Infection Biology in Berlin to the NCBI SRA database under the BioProject [PRJNA1401910](https://www.ncbi.nlm.nih.gov/bioproject/PRJNA1401910). Each of our partnering institutions is planning to publish a manuscript presenting their population genetic and epidemiological findings, at which point they will publicly release their raw sequencing data.

4. It is advisable to further analyze whether there are differences in detection accuracy and sensitivity between nanopore sequencing and mainstream methods such as Illumina sequencing.

In the discussion we highlight that Illumina sequencing approaches will have some advantages over longer-read nanopore approaches, in particular with relation to sensitivity. However, as outlined in our introductory remarks, while a direct comparison between our protocol and an Illumina-based protocol (e.g. WGS, MadHatter, Pf-SMARRT) would be interesting, it falls outside of the scope of our current manuscript. The focus of our manuscript is method development and validation, which is achieved by comparison against a known truth set. We used a panel of mock laboratory strains as our truth set, which is a gold-standard and aligns with other publications in which nanopore (e.g. <https://journals.plos.org/globalpublichealth/article?id=10.1371/journal.pgph.0002743>) or Illumina-based methods (e.g. <https://www.medrxiv.org/content/10.1101/2024.10.03.24314715v2>; <https://www.nature.com/articles/s41598-025-94716-5>) were developed and validated. This is distinct from method comparison (e.g. ONT vs Illumina), which looks at strengths / weaknesses / differences of various approaches. Typically, method comparison is done in stand-alone publications which include, for fairness, authors who co-developed the different methods being compared (for e.g. see the recent MIPs versus MadHatter paper: <https://www.researchsquare.com/article/rs-5743980/v1>). In addition, please appreciate that conducting such a comparative analysis is a challenging and expensive undertaking, as the same set of samples needs to be processed on two platforms; and it requires expertise with both Illumina and ONT (for lab and bioinformatic analysis), which our current team does not have.

5. It's suggested to clarify the typical Plasmodium density in patients under normal circumstances, explain how the threshold of 100 parasites/ μ L (mentioned in the manuscript) is defined, and note that this threshold seems relatively high.

In the results we now more explicitly describe our assay cutoff (100-1000p/uL), while highlighting that this may vary due to sample DNA quality, and should be established on a per study basis.

Unfortunately providing a typical parasite density in patients is not possible, as it is influenced by a wide array of factors including age, immunity level, parasite genetic factors, co-infection status, etc. As a result, what is typical varies dramatically depending on the study population (e.g. from 100 to >10,000 parasites/uL). In Fig. 2a we show the distributions of parasite densities for 950 field samples from seven countries in sub-Saharan Africa, which we would regard as typical, and the median parasitemia as 3,645 parasites/uL (IQR 621 – 11,540). For therapeutic efficacy studies (TES) in which clinical patients are enrolled, the WHO has the following inclusion criteria: >2,000p/uL for high transmission zones, >1,000p/uL for moderate transmission zones, and >250p/uL for low transmission zones (<https://iris.who.int/handle/10665/272284>; see section 4.2.1); these values would represent lower limits for clinical cases, and we note that they are all higher than our assays cut-offs. A parasitemia of 100p/uL is also lower than standard RDTs (~200p/uL), which implies that our assay would work well for any study which recruits patients using RDTs.

We highlight in the Discussion that short-read approaches can have higher sensitivities than our assay, although we note that at our current sensitivity, our assay can be used productively across a wide range of MMS activities; this is also highlighted by Reviewer #1.